# PRMT3 interacts with ALDH1A1 and regulates gene-expression by inhibiting retinoic acid signaling

Mamta Verma [1,5], Mohd. Imran K. Khan[1,5], Rajashekar Varma Kadumuri[2], Baskar Chakrapani[1], Sharad Awasthi[1], Arun Mahesh[1], Gayathri Govindaraju[3], Pavithra L Chavali[4], Arumugam Rajavelu[3], Sreenivas Chavali [2✉] & Arunkumar Dhayalan [1✉]

Protein arginine methyltransferase 3 (PRMT3) regulates protein functions by introducing asymmetric dimethylation marks at the arginine residues in proteins. However, very little is known about the interaction partners of PRMT3 and their functional outcomes. Using yeast-two hybrid screening, we identified Retinal dehydrogenase 1 (ALDH1A1) as a potential interaction partner of PRMT3 and confirmed this interaction using different methods. ALDH1A1 regulates variety of cellular processes by catalyzing the conversion of retinaldehyde to retinoic acid. By molecular docking and site-directed mutagenesis, we identified the specific residues in the catalytic domain of PRMT3 that facilitate interaction with the C-terminal region of ALDH1A1. PRMT3 inhibits the enzymatic activity of ALDH1A1 and negatively regulates the expression of retinoic acid responsive genes in a methyltransferase activity independent manner. Our findings show that in addition to regulating protein functions by introducing methylation modifications, PRMT3 could also regulate global gene expression through protein-protein interactions.

[1] Department of Biotechnology, Pondicherry University, Puducherry 605014, India. [2] Department of Biology, Indian Institute of Science Education and Research (IISER) Tirupati, Tirupati, Andhra Pradesh 517507, India. [3] Interdisciplinary Biology, Rajiv Gandhi Centre for Biotechnology, Trivandrum, Kerala 695014, India. [4] CSIR-Centre for Cellular & Molecular Biology, Hyderabad, Telangana 500007, India. [5]These authors contributed equally: Mamta Verma, Mohd. Imran K. Khan. ✉email: schavali@iisertirupati.ac.in; arun.dbt@pondiuni.edu.in

 1

Protein arginine methyltransferase 3 (PRMT3) is a member of the type I protein arginine methyltransferase family. PRMT3 catalyzes the asymmetric dimethylation modification of the arginine residues, in diverse substrate proteins in vitro such as GST-GAR, PABPII, hnRNPA1, HMGA1, and PABPN1[1–5]. PRMT3 interacts with the ribosomal small subunit protein 2 (rPS2) and methylates it, both in vitro and in vivo[6,7]. The rpS2 interaction with the PRMT3 promotes the PRMT3 association to ribosomes and increases the stability of rpS2 by reducing the ubiquitination of rpS2[6,8]. However, the precise functional significance of this interaction and methylation is not known, since the levels of 40S, 60S, 80S monosomes, and polysomes are unaltered in PRMT3-deficient mice[7]. Depletion of PRMT3 in the rat hippocampal neurons leads to a decrease in the stability of rpS2 and defective spine formation in the neurons[9]. In contrast to other PRMTs, PRMT3 is predominantly cytoplasmic and possesses a C2H2 zinc finger at the N-terminal region, which is required for its optimal enzymatic activity and contributes to the in vivo substrate specificity[1,3,10]. Palmitic acid (PA) treatment induces the translocation of PRMT3 to the nucleus and promotes the expression of lipogenic proteins through LXRα interaction[11]. The myokine irisin prevents the PA-induced lipogenesis and oxidative stress by reducing the PRMT3 levels[12].

While few substrates of PRMT3 have been identified, very few protein–protein interactions of PRMT3 are comprehensively functionally characterized. For instance, the tumor-suppressor protein, DAL-1 has been shown to interact with PRMT3 and inhibit its activity[13]. In this study, we aimed to characterize the interaction partners of PRMT3 and study the functional outcomes of the interaction(s). For this, we performed yeast two-hybrid screening of PRMT3 with human cDNA library and identified aldehyde dehydrogenase 1 family member A1 (ALDH1A1) as a potential interaction partner of PRMT3.

ALDH1A1 is the major enzyme that catalyzes the synthesis of retinoic acid (RA) from retinaldehyde (RALD) and hence plays a major role in retinoid (vitamin A) signaling[14]. ALDH1A1 is a functionally versatile enzyme that participates in diverse processes. For instance, ALDH1A1 is involved in the detoxification of anticancer drugs, aliphatic aldehydes, products of lipid peroxidation, a product of protein deglycation, the DOPAL in dopaminergic neurons, and in the prevention of ultraviolet radiation-induced damages in the ocular tissues[15–24]. ALDH1A1 is the key enzyme involved in the synthesis of γ-aminobutyric acid (GABA) in dopamine neurons[25]. ALDH1A1 knockout mice resist the high fat-induced insulin resistance and obesity and display increased metabolic rates; reduced fasting glucose levels by the inhibition of gluconeogenesis and reduced hepatic triacylglycerol synthesis[26–29]. High expression and activity of ALDH1A1 is a characteristic feature of stem cells of normal tissues and cancer stem cells and this property has been exploited for the isolation of stem cells. Elevated levels of ALDH1A1 is often associated with poor clinical outcomes of various cancers[30,31]. Here, we report that PRMT3 interacts with ALDH1A1 and inhibits its activity, leading to negative regulation of the expression of retinoic acid-responsive genes.

## Results

**PRMT3 interacts with ALDH1A1**. We performed yeast two-hybrid screening using human PRMT3 as bait, with a universal human normalized cDNA library, which also includes low copy number transcripts, to identify novel interaction partners. The stringent screening resulted in two positive clones. Sequencing revealed that both the clones contain the C-terminal region (amino acid residues 371–501) of the Retinal dehydrogenase 1 (ALDH1A1) protein in frame with GAL4 activation domain. The

observed interaction was confirmed with the full-length ALDH1A1 in yeast two-hybrid assay (Fig. 1a). To validate the observed interaction in mammalian cells, we performed (i) co-immunoprecipitation (Co-IP) experiments using GFP-tagged PRMT3 and the Myc-tagged ALDH1A1 constructs, and (ii) reverse Co-IP experiments, using the GFP-tagged ALDH1A1 and HA-tagged PRMT3 constructs in HEK293 cells. We observed that PRMT3 efficiently precipitated the ALDH1A1 protein and vice versa, indicating that the PRMT3 could interact with ALDH1A1 in mammalian cells (Fig. 1b). We next checked whether PRMT3–ALDH1A1 interaction occurs endogenously in cells, i.e., at physiological levels of both the proteins. To study this, we performed immunoprecipitation experiments in HEK293 cells using the ALDH1A1-specific antibody. We found that ALDH1A1 efficiently precipitated the PRMT3 enzyme. Importantly, control IgG failed to immunoprecipitate PRMT3, indicating the specificity of immunoprecipitation experiments. These results confirm that PRMT3 interacts with ALDH1A1 at their endogenous levels (Fig. 1c). To investigate if the observed interaction results from direct binding, we performed the GST and Ni-NTA pulldown assays using the affinity-purified GST-tagged PRMT3 and His-tagged ALDH1A1. We found that the PRMT3 interacted directly with ALDH1A1 (Fig. 1d, e). We further substantiated these findings by confocal microscopy using the GFP-tagged PRMT3 and DsRed-tagged ALDH1A1 constructs in HEK293 cells. Both GFP-PRMT3 and DsRed-ALDH1A1 displayed a strong cytoplasmic localization pattern, corroborating the possibility of this interaction in vivo (Supplementary Fig. 1). Collectively, these findings establish that the PRMT3 interacts with ALDH1A1.

**The catalytic domain of PRMT3 interacts with the C-terminal region of ALDH1A1**. Next, we sought to map the interacting regions of PRMT3 and ALDH1A1. The N-terminal region of PRMT3 contains a zinc finger domain, which is known to confer the in vivo substrate specificity, while the C-terminal catalytic domain brings about substrate methylation[10]. To identify the ALDH1A1 interacting regions of PRMT3, we cloned (i) the N-terminal region of PRMT3 (PR-NTR; residues 1–186) encompassing the zinc finger domain, (ii) the C-terminal region (PR-CTR; residues 186–531), and (iii) full-length PRMT3 (PR-FL) into the mammalian expression construct in frame with GFP. Likewise, for ALDH1A1, we cloned (i) the N-terminal region (AL-NTR; residues 1–335), (ii) the C-terminal region (AL-CTR; residues 336–501), and (iii) the full-length ALDH1A1 (AL-FL) in bacterial expression cassette in fusion with GST. Using GST-tagged ALDH1A1 as bait, we performed pull-down experiments with HEK293 cell lysates overexpressing GFP-tagged PRMT3 full-length or truncated proteins. The GST pull-down assays showed the interaction between the full-length proteins as expected (Fig. 1f). Among the truncated proteins, PR-CTR showed interaction with AL-CTR, comparable to that of full-length proteins, while we could not detect any interaction with that of the N-terminal regions of either protein (Fig. 1f). Importantly, evolutionary analyses indicate that the C-terminal region of ALDH1A1 that interacts with PRMT3 might have coevolved with PRMT3, providing another important line of evidence for the importance of this region in ALDH1A1 in facilitating PRMT3–ALDH1A1 interaction (Supplementary Fig. 2).

The C-terminal region of PRMT3 which interacts with ALDH1A1 also harbors the catalytic domain that catalyzes the methyltransferase activity of PRMT3. This prompted us to investigate whether the association of PRMT3 with the methyl group donor, S-adenosyl-L-methionine (SAM), or the resulting product upon methyl transfer, S-adenosyl-L-homocysteine (SAH), affects the PRMT3–ALDH1A1 interaction. For this, we

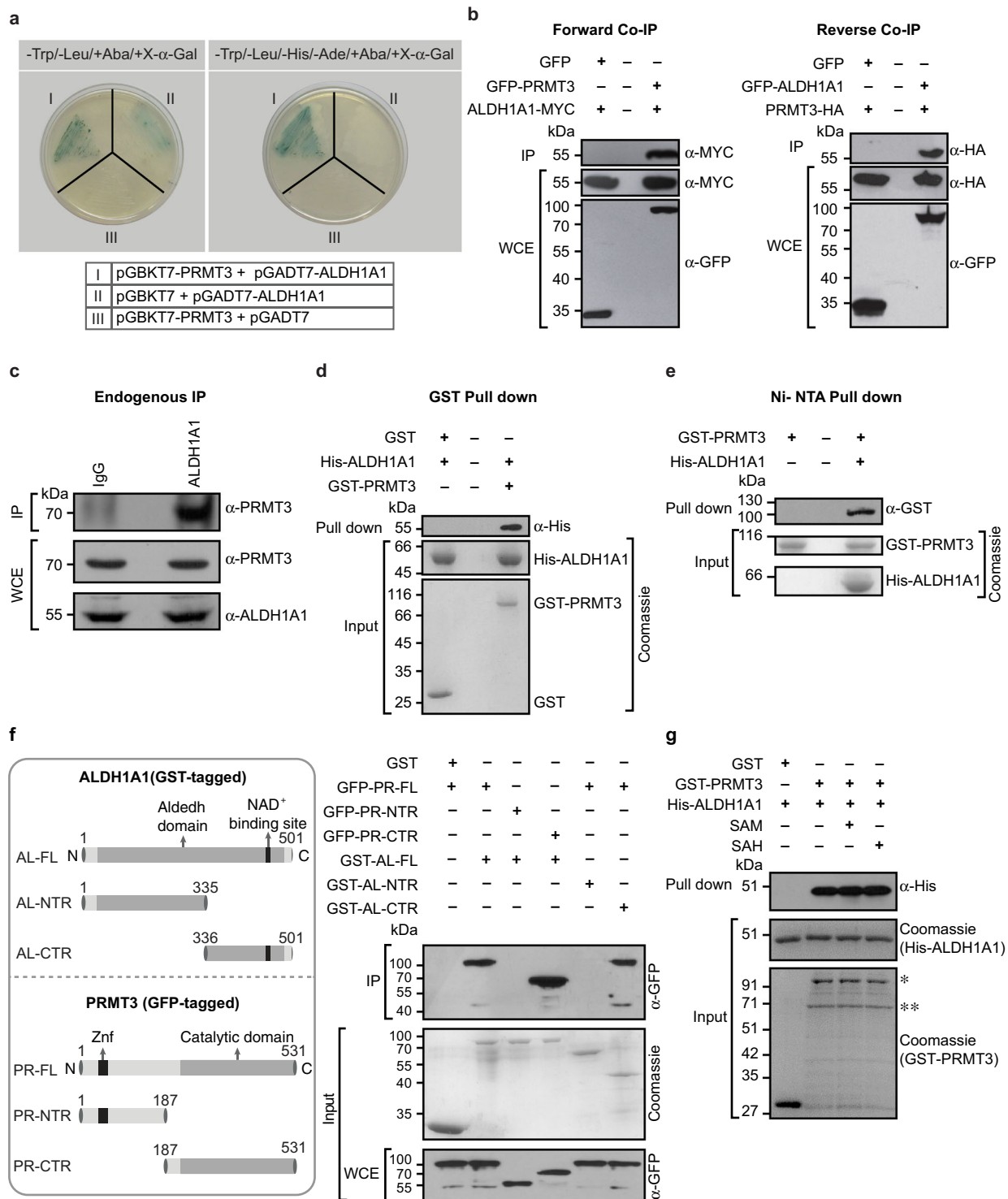

tested the effect of SAM or SAH addition on PRMT3–ALDH1A1 interaction by performing GST pull-down assays. We found that the addition of SAM or SAH did not affect the PRMT3–ALDH1A1 interaction (Fig. 1g). Taken together, these findings establish that the C-terminal catalytic domain of PRMT3 interacts with the C-terminal region of ALDH1A1.

**PRMT3 exhibits strong binding with ALDH1A1.** Human protein arginine methyltransferase (PRMT) family consists of nine members which are classified into Type I (PRMT1– PRMT4, PRMT6, and PRMT8), Type II (PRMT5 and PRMT9), and Type

III (PRMT7), based on the type of methylarginine residues they generate[32–34] (Supplementary Table 1). Since the catalytic domains of PRMTs and aldehyde dehydrogenases (ALDHs) are conserved, we next tested whether ALDH1A1 could interact with the other PRMT family members and likewise if PRMT3 could interact with the other ALDH family members. For this, we selected representatives from each type of PRMTs (Type I: PRMT2 and PRMT6; Type II: PRMT5; and Type III: PRMT7; Fig. 2a). We then tested the interaction of these representative PRMTs with ALDH1A1 through GST pull-down assays, using PRMT3 as a positive control. We could not detect any interaction of ALDH1A1 with PRMT2,

**Fig. 1 PRMT3 interacts with ALDH1A1. a** Yeast two-hybrid assay to investigate the interaction of the full-length PRMT3 and the full-length ALDH1A1. We scored the interaction by profiling the expression of two or four reporter genes, as indicated. **b** PRMT3 interacts with ALDH1A1 in mammalian cells. Left panel represents the results of forward Co-IP. The co-immunoprecipitation was performed in HEK293 cells co-expressing GFP or GFP-PRMT3 and ALDH1A1-Myc. The immunoprecipitated samples were analyzed by western blotting using Myc antibody (upper blot). The whole-cell extracts were probed with Myc antibody (middle blot) and GFP antibody (lower blot). Right panel represents the results of reverse Co-IP. The co-immunoprecipitation was performed in HEK293 cells co-expressing GFP or GFP-ALDH1A1 and PRMT3-HA. The immunoprecipitated samples were analyzed by western blotting using HA antibody (upper blot). The whole-cell extracts were probed with HA antibody (middle blot) and GFP antibody (lower blot). IP immunoprecipitation, WCE whole-cell extract. **c** PRMT3 interacts with ALDH1A1 endogenously. Immunoprecipitation was performed in HEK293 cells using ALDH1A1 antibody or control IgG and the bound fractions were probed with PRMT3 antibody (upper blot). The whole-cell extracts were probed with PRMT3 antibody (middle blot) and ALDH1A1 antibody (lower blot). **d** GST pull-down assay shows the direct interaction of PRMT3 and ALDH1A1. The GST-tagged PRMT3 or GST protein was coupled to glutathione sepharose, and the beads were incubated with His-tagged ALDH1A1. The bound fractions were immunoblotted with His antibody (upper blot). About 4% of His-tagged ALDH1A1 (middle blot) or 2.5% of GST and GST-tagged PRMT3 (lower blot), used in the pull-down assay, were separated in SDS-PAGE and stained with coomassie blue dye. **e** Ni-NTA pull-down assay shows the direct interaction of PRMT3 and ALDH1A1. The GST-tagged PRMT3 was incubated with Ni-NTA beads or Ni-NTA beads, which were coupled with His-tagged ALDH1A1. The bound fractions were separated and immunoblotted with GST antibody (upper blot). About 2.5% of GST-PRMT3 (middle blot) and 4% of His-tagged ALDH1A1 (lower blot), used in the pull-down assay, were separated in SDS-PAGE and stained with coomassie blue dye. **f** The catalytic domain of PRMT3 interacts with the C-terminal region of ALDH1A1. The GST-tagged full-length ALDH1A1 (AL-FL) or N-terminal region of ALDH1A1 (AL-NTR) or C-terminal region of ALDH1A1 (AL-CTR) or the GST protein were coupled to the glutathione sepharose, and the beads were incubated with the cell lysates prepared from the cells, which were overexpressing of GFP-tagged PRMT3 full-length protein (PR-FL) or GFP-tagged N-terminal region of PRMT3 protein (PR-NTR) or GFP-tagged C-terminal catalytic domain of PRMT3 protein (PR-CTR). The bound fractions were immunoblotted with GFP antibody (upper blot). About 8% of the GST or GST-tagged ALDH1A1 full length or truncated proteins, which were used in pull-down assay were separated in SDS-PAGE and stained with coomassie blue dye (middle blot) and 2% of the whole-cell lysates used in the pull-down assay, were probed with GFP antibody (lower blot). **g** GST pull-down assay shows that the interaction of PRMT3 and ALDH1A1 is not affected by the addition of SAM or SAH. The GST-tagged PRMT3 or GST protein was coupled to glutathione sepharose, and the beads were incubated with His-tagged ALDH1A1 with or without 200 μM SAM or SAH. The bound fractions were immunoblotted with His antibody (upper blot). About 2% of His-tagged ALDH1A1 (middle blot) or 2.5% of GST and GST-tagged PRMT3 (lower blot), used in the pull-down assay, were separated in SDS-PAGE and stained with coomassie blue dye. *Indicates the full-length GST-PRMT3, and **indicates the degraded GST-PRMT3 (Supplementary Fig. 10).

PRMT5, and PRMT7, while ALDH1A1 showed a feeble interaction with PRMT6, which was very weak compared to the PRMT3–ALDH1A1 interaction (Fig. 2b).

For studying the interaction of other ALDHs with PRMT3, we first generated a phylogenetic tree of human ALDH members (Supplementary Table 1) by performing multiple sequence alignment. Based on the phylogenetic tree, we selected the other two ALDH1 members (ALDH1A2 and ALDH1A3), ALDH2 which is a closely related member to the ALDH1 clade, and ALDH3A1 as a distantly related outgroup member (Fig. 2c). We then performed GST pull-down experiments to test the interactions of ALDH1A2, ALDH1A3, ALDH2, and ALDH3A1 with PRMT3, using ALDH1A1 as a positive control. Interestingly, PRMT3 showed weak interactions with ALDH1A2, ALDH1A3, and ALDH3A1, while we could not detect interaction with ALDH2 (Fig. 2d). Notably, these interactions were much weaker compared to PRMT3–ALDH1A1. These findings establish that among the tested PRMT and ALDH members, PRMT3 exhibits the strongest interaction with ALDH1A1. Based on these results, we focused on PRMT3–ALDH1A1 interaction for further studies.

**The C-terminal residues that lie outside the catalytic active site of PRMT3 facilitate its interaction with ALDH1A1.** To obtain structural insights into the PRMT3–ALDH1A1 interaction, we performed global molecular docking of human PRMT3 (PDB ID: 2FYT) with human ALDH1A1 (PDB ID: 5L2M) crystal structures using HawkDock server coupled with Molecular Mechanics/Generalized Born Surface Area (MM/GBSA) analysis to identify key residues involved in the interaction (Fig. 3a, b). We observed that PRMT3 forms a strong interaction complex with ALDH1A1 with a calculated binding free energy of −15.70 kcal/mol. MM/GBSA free energy calculations revealed that His464, Asn465, Arg466, and Val468 of PRMT3 to be important facilitators of this interaction (Fig. 3a, b). From the MM/GBSA calculations, we observed that (i) the side chain of Arg466 of PRMT3 contributes to strong electrostatic interaction with the side chain of Asp317 of

ALDH1A1, (ii) Val468 of PRMT3 is involved in hydrophobic interaction with a valine hydrophobic cluster (Val320, Val324, Val386) of ALDH1A1, (iii) His464 of PRMT3 might contribute to partial cation–π interaction with Lys410 of ALDH1A1, and (iv) the side chain of Asn465 of PRMT3 could form a polar contact with the side chain of Lys410 (Fig. 3a, b), all of which contribute to the PRMT3–ALDH1A1 interaction.

To determine the importance of the residues identified from the molecular docking studies, we mutated each of the His464, Asn465, Arg466, and Val468 residues of PRMT3 located in the interaction interface to alanine. We investigated the interaction of these PRMT3 mutant proteins with ALDH1A1 through GST pull-down experiments. We found that these mutations drastically reduced the PRMT3–ALDH1A1 interaction indicating the reliability of the modeled PRMT3–ALDH1A1 complex structure (Fig. 3c). Collectively, these results suggest that the PRMT3–ALDH1A1 interaction is facilitated by the residues that lie outside the catalytic active site in the C-terminal region of PRMT3.

**PRMT3 inhibits the enzymatic activity of ALDH1A1.** Given that, ALDH1A1 catalyzes the irreversible oxidation of a variety of cellular aldehydes[35], we next set out to understand the impact of PRMT3–ALDH1A1 interaction on the enzymatic activity of ALDH1A1. For this, we measured the enzymatic activity of ALDH1A1 in vitro using propionaldehyde as substrate and NAD + as a cofactor by quantifying the formation of NADH over time in the presence or absence of full length (PR-FL) or truncated versions of PRMT3 (PR-NTR and PR-CTR). Strikingly, we found that PR-FL and PR-CTR inhibited the ALDH1A1 activity significantly (Fig. 4a, b). However, we could not detect inhibition of ALDH1A1 enzymatic activity by the PR-NTR. Importantly, PRMT3 inhibited ALDH1A1 enzymatic activity in a concentration-dependent manner (Supplementary Fig. 3). Given that the C-terminal catalytic domain of PRMT3 interacts with ALDH1A1 (Fig. 1f) and that the C-terminal region was alone

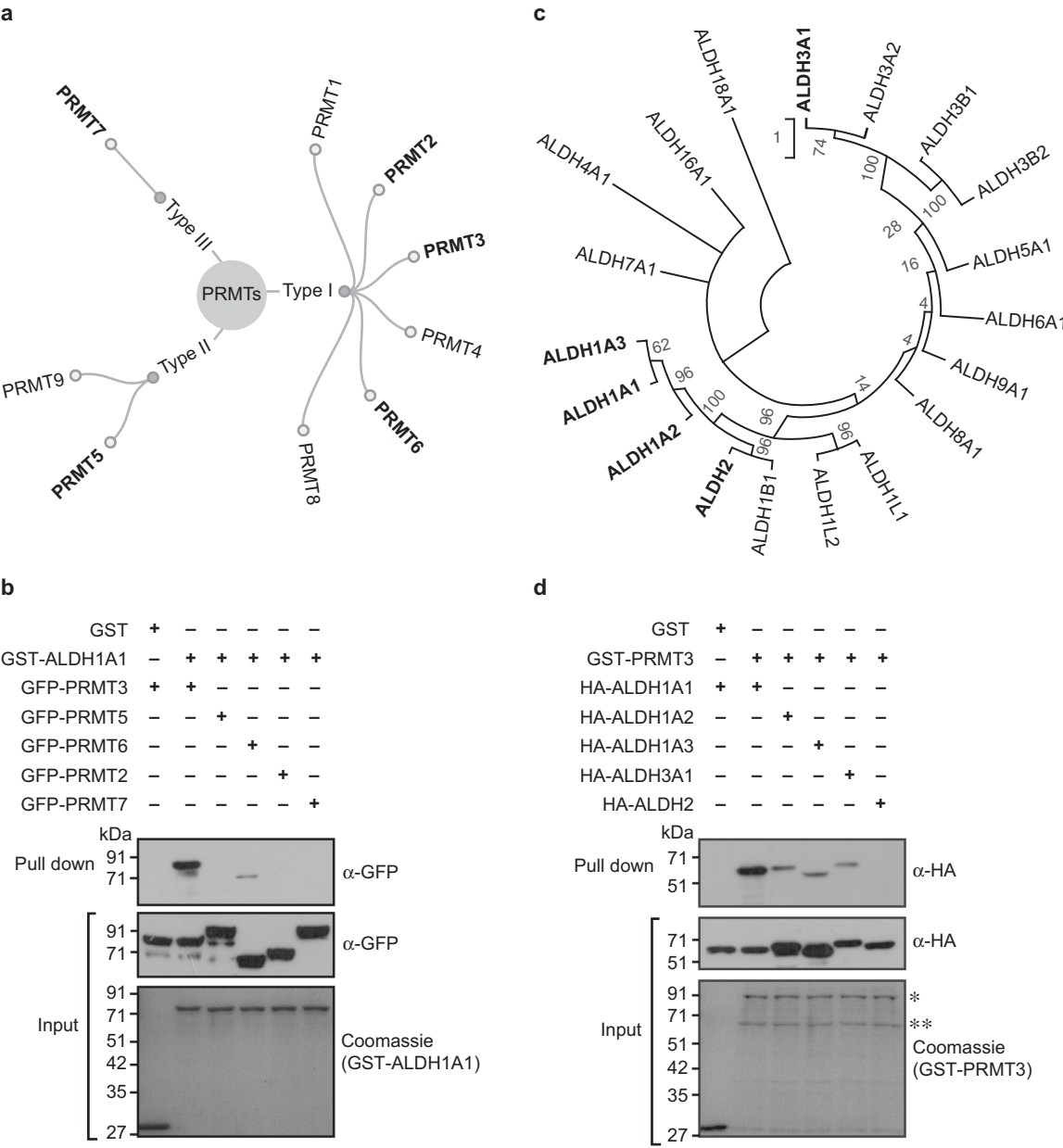

**Fig. 2 PRMT3 exhibits strong binding with ALDH1A1. a** Based on the type of methylarginine that they generate, different members of protein arginine methyltransferases are classified into type I (that generate asymmetric dimethylarginine), type II (that generate symmetric dimethylarginine), and type III (that generate monomethyl arginine). **b** ALDH1A1 interacts with PRMT3 strongly. The GST-tagged ALDH1A1 or GST protein was coupled to glutathione sepharose, and the beads were incubated with the whole-cell extracts prepared from HEK293 cells overexpressing GFP-PRMT3 or GFP-PRMT2 or GFP-PRMT5 or GFP-PRMT6 or GFP-PRMT7. The bound fractions were immunoblotted with GFP antibody (upper blot). The whole-cell extracts were probed with GFP antibody (middle blot). About 2.5% of GST and GST-tagged ALDH1A1 (lower blot), used in the pull-down assay, were separated in SDS-PAGE and stained with coomassie blue dye. **c** Phylogenetic tree showing the relationship among different human ALDH family of enzymes, obtained using multiple sequence alignment of the protein sequences. Multiple sequence alignment was generated using MAFFT[63] and the phylogenetic tree was reconstructed using MEGA-X[76]. **d** PRMT3 interacts with ALDH1A1 strongly. The GST-tagged PRMT3 or GST protein was coupled to glutathione sepharose and the beads were incubated with the whole-cell extracts prepared from HEK293 cells overexpressing HA-ALDH1A1 or HA-ALDH1A2 or HA-ALDH1A3 or HA-ALDH3A1 or HA-ALDH2. The bound fractions were immunoblotted with HA antibody (upper blot). The whole-cell extracts were probed with HA antibody (middle blot). About 2.5% of GST and GST-tagged PRMT3 (lower blot), used in the pull-down assay, were separated in SDS-PAGE and stained with coomassie blue dye. *Indicates the full-length GST-PRMT3, and **indicates the degraded GST-PRMT3 (Supplementary Fig. 10).

sufficient to inhibit the activity of ALDH1A1, we conclude that the interaction of PRMT3 could inhibit the enzymatic activity of ALDH1A1. Since we did not include the methylation cofactor S-adenosyl-L-methionine (SAM) in these enzymatic assays, it is highly unlikely that the PRMT3-mediated methylation of ALDH1A1 might inhibit the enzymatic activity of ALDH1A1.

Nevertheless, to probe if ALDH1A1 is methylated by PRMT3, we performed in vitro methylation assay using radiolabelled SAM. While the positive control GST-GAR was efficiently methylated by PRMT3, we could not detect methylation of ALDH1A1 by PRMT3 (Supplementary Fig. 4). Therefore, though the C-terminal region of PRMT3 containing the catalytic domain

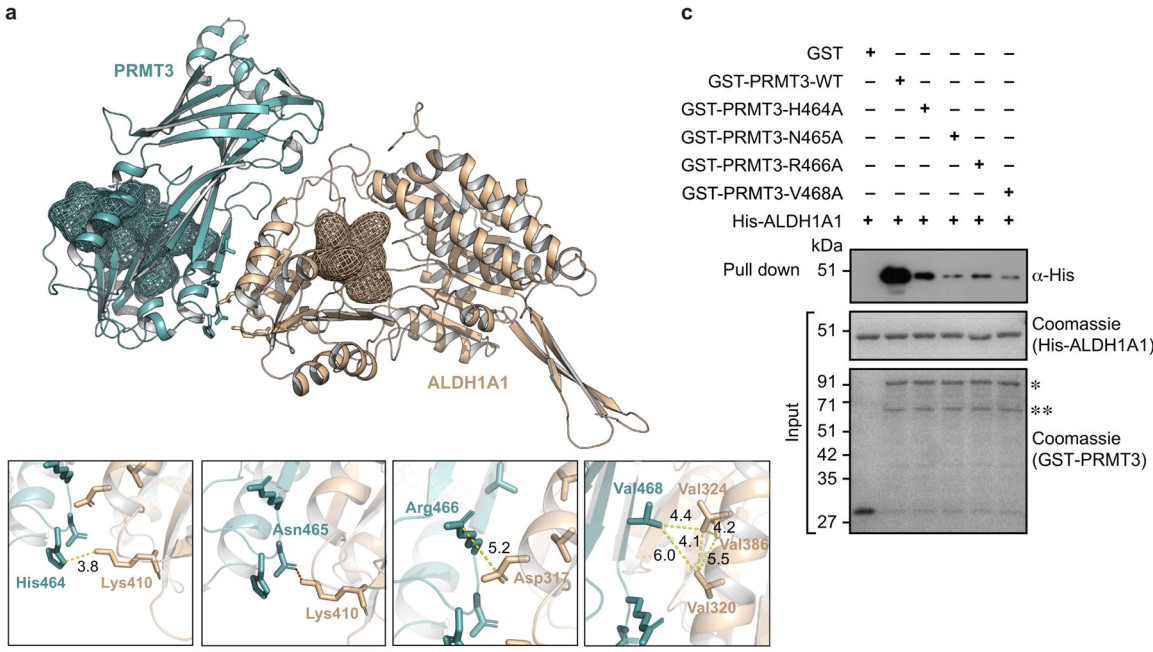

**b**

| PRMT3 residue | Van der waals energy (kcal/mol) | Electrostatic energy (kcal/mol) | Generalized Born solvation energy (kcal/mol) | Solvent accessible solvation energy (kcal/mol) | Total binding energy (kcal/mol) |
|---|---|---|---|---|---|
| His464 | −2.05 | −8.82 | 7.54 | −0.36 | −3.69 |
| Asn465 | −3.2 | −10.35 | 10.7 | −0.43 | −3.28 |
| Arg466 | −1.75 | −38.96 | 37.5 | −0.5 | −3.7 |
| Val468 | −3.13 | 3.47 | −2.2 | −0.57 | −2.43 |

**Fig. 3 C-terminal region residues that lie outside the PRMT3 catalytic domain facilitate interaction with ALDH1A1. a** Crystal structure of human PRMT3 (cyan) in complex with human ALDH1A1 (brown). Catalytically active site regions of PRMT3 and ALDH1A1 are represented in the mesh. Key residue—interactions between PRMT3 (cyan) and ALDH1A1 (brown) are shown as insets of enlarged view. Distance (in Angstroms) between residues are given for electrostatic and hydrophobic interactions in a yellow dotted line. Polar contacts are given in the red dotted line. **b** Molecular Mechanics/Generalized Born Surface Area (MM/GBSA) binding free energies of the PRMT3 residues—His464, Asn465, Arg466, and Val468 that are predicted to be involved in the interaction with ALDH1A1. **c** Interaction profiles of the wild-type PRMT3 and predicted interaction interface alanine mutants of PRMT3 with ALDH1A1. The GST or GST-PRMT3 wild-type or GST-PRMT3-H464A or GST-PRMT3-N465A or GST-PRMT3-R466A or GST-PRMT3-V468A mutant proteins were coupled to glutathione sepharose and the beads were incubated with His-tagged ALDH1A1. The bound fractions were immunoblotted with His antibody (upper blot). About 2% of His-tagged ALDH1A1 (middle blot) or 2.5% of GST and GST-PRMT3 wild-type and mutant proteins (lower blot), used in the pull-down assay, were separated in SDS-PAGE and stained with coomassie blue dye. *Indicates the full-length GST-PRMT3, and **indicates the degraded GST-PRMT3 (Supplementary Fig. 10).

facilitates the interaction with ALDH1A1, PRMT3 does not methylate ALDH1A1. Collectively these data establish that the direct interaction of PRMT3 with ALDH1A1 inhibits the enzymatic activity of ALDH1A1.

These observations prompted us to investigate whether PRMT3 could inhibit ALDH1A1 activity in vivo. For this, we perturbed the levels of PRMT3 in HEK293 cells through siRNA-mediated depletion or by overexpression and quantified the ALDH1A1 activity in these cells using ALDEFLUOR assay. ALDEFLUOR assay measures the in vivo enzymatic activity of aldehyde dehydrogenases by using a nontoxic fluorescent substrate. The efficient knockdown of PRMT3 was confirmed by quantitative RT-PCR and western blotting (Supplementary Fig. 5 and Fig. 5a). The overexpression of HA-tagged PRMT3 increased the protein levels of PRMT3 by ~4.2-fold compared to the endogenous levels of PRMT3 (Fig. 5a). We found that cells overexpressing PRMT3 exhibited a significant reduction (~27%) in ALDEFLUOR fluorescence compared to the vector-transfected control cells. Consistently, PRMT3 depletion led to a significant increase (~24%) in the ALDEFLUOR fluorescence (Fig. 5a). These findings suggest that the levels of PRMT3 are inversely

correlated to the enzymatic activity of ALDH1A1 in vivo. PRMT3 might regulate the activity of ALDH1A1 in three possible ways. PRMT3 binds to ALDH1A1 and (i) alters the steady-state level of ALDH1A1, or (ii) affects the enzymatic activity of ALDH1A1, or (iii) both. To gain a mechanistic understanding of this, we perturbed the levels of PRMT3 in HEK293 cells and quantified the levels of ALDH1A1 both at transcripts and protein levels. We found that the perturbation of PRMT3 did not alter the transcript (Supplementary Fig. 6a) and protein levels of ALDH1A1 (Supplementary Fig. 6b). This suggests that PRMT3 inhibits the enzymatic activity of ALDH1A1 without affecting the abundance of ALDH1A1. Taken together, these findings demonstrate that PRMT3 inhibits the enzymatic activity of ALDH1A1 in vivo.

**PRMT3 negatively regulates ALDH1A1-mediated retinoic acid signaling.** ALDH1A1 is the major enzyme that catalyzes the synthesis of retinoic acid (RA) from retinaldehyde (RALD) and hence is a major determinant of retinoid signaling and downstream functional outcomes[14]. Our finding that PRMT3 binding could inhibit ALDH1A1 enzymatic activity prompted us to probe the impact of this interaction on RA signaling. For this, we

**a**

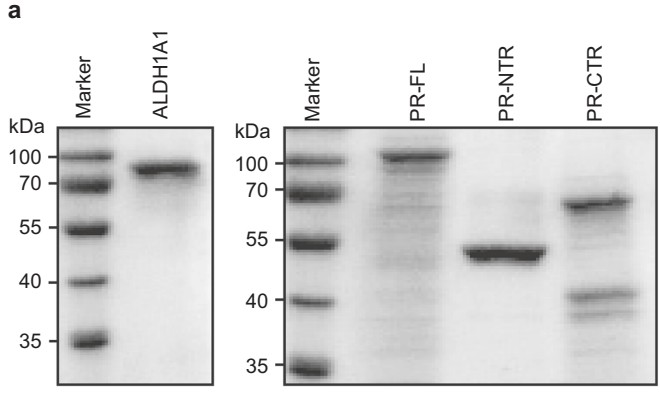

**b**

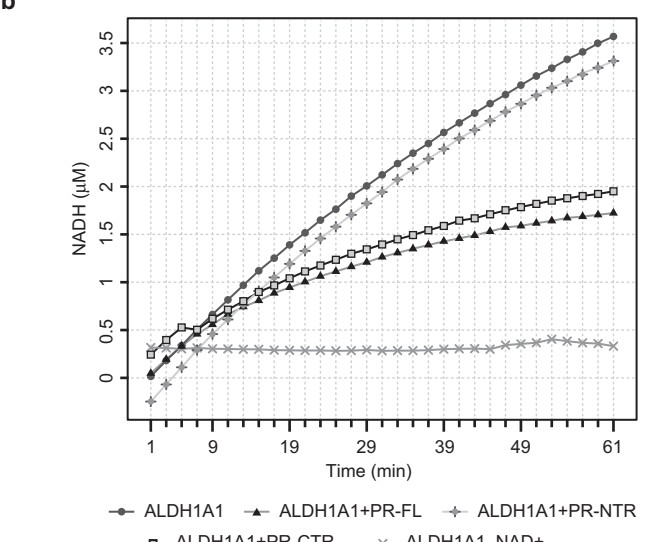

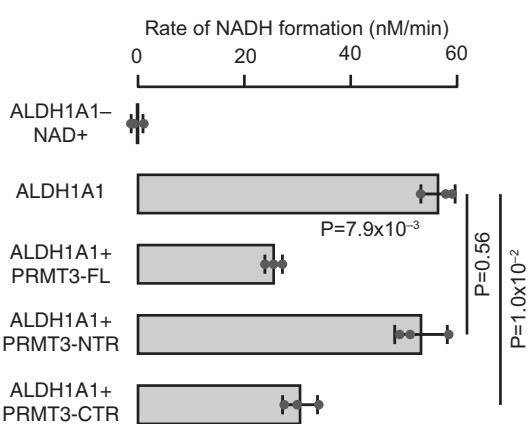

**Fig. 4 PRMT3 inhibits the ALDH1A1 activity in vitro. a** Purification of GST-tagged ALDH1A1-FL, GST-tagged PRMT3 full-length protein, GST-tagged N-terminal region of PRMT, and GST-tagged catalytic domain of PRMT3 proteins. Coomassie-stained gel of purified GST-tagged ALDH1A1-FL (left panel) and GST-tagged PRMT3 full-length protein (PR-FL), GST-tagged N-terminal region of PRMT3 (PR-NTR), and GST-tagged catalytic domain of PRMT3 proteins (PR-CTR) (right panel). **b** PRMT3 inhibits the ALDH1A1 activity in vitro. ALDH1A1 activity was measured in the presence and absence of GST-tagged PRMT3 full-length protein or GST-tagged N-terminal region of PRMT3 or GST-tagged catalytic domain of PRMT3. The representative activity of the ALDH1A1 of the three independent experiments is shown in the graph (top panel), and the mean initial slopes of the three independent reactions were plotted as the rate of the reactions (bottom panel). Error bar indicates the standard deviations of the mean. The statistical significance was assessed by a two-tailed $t$ test (Supplementary Fig. 10 and Supplementary Data 1).

hence an increase in RARE-dependent gene expression. Using this expression system, we investigated the effect of PRMT3 perturbation on the luciferase activity. The efficiency of PRMT3 perturbation in these samples was quantified by immunoblotting with PRMT3 antibody (Fig. 5b, upper panel blots). We found that the overexpression of wild-type PRMT3 or catalytically inactive mutant (E338Q) of PRMT3[36–38] reduced the reporter luciferase activity significantly (~30%) compared to the vector control cells (Fig. 5b). We did not observe any significant difference between wild-type PRMT3 and PRMT3-E338Q mutant on the magnitude of reduction of luciferase activity, indicating that PRMT3 negatively regulates RA signaling in a methyltransferase activity-independent manner. Importantly, PRMT3 depletion led to a significant increase (~24%) in the RARE-luciferase activity compared to the control siRNA-treated cells (Fig. 5b). These results show that ALDH1A1-mediated conversion of RALD to RA is inhibited by PRMT3.

To examine the impact of PRMT3–ALDH1A1 interaction on gene-expression regulation by RA in vivo, we measured the mRNA levels of RA-responsive genes[39–45] involved in diverse biological processes (Supplementary Table 2) (i) with or without overexpressing wild-type PRMT3 or PRMT3-E338Q mutant, (ii) with control siRNA or PRMT3 specific siRNA and (iii) in the presence or absence of RALD using quantitative RT-PCR (Fig. 6 and Supplementary Fig. 9). The efficiency of PRMT3 perturbation in these samples was quantified by immunoblotting with PRMT3 antibody (Supplementary Figs. 8a, b, 9a). We found that the overexpression of wild-type PRMT3 or the catalytically inactive mutant PRMT3-E338Q significantly decreased the expression of all the tested RA-responsive genes to varying extents in the HEK293 cells treated with RALD (Fig. 6 and Supplementary Fig. 9b). Importantly, there was no significant difference in the impact of the wild-type PRMT3 and PRMT3-E338Q mutant on the expression of tested RA-responsive transcripts. This indicates that PRMT3 represses the expression of RA target genes in a methyltransferase activity-independent manner (Supplementary Fig. 9). Strikingly, PRMT3 overexpression resulted in inconsistent effects on the levels of tested RA-responsive transcripts ranging from no changes to mild increase or decrease in the absence of RALD treatment (Fig. 6). These results suggest that PRMT3 reduces the expression of RA-responsive genes in a RALD-dependent manner. Consistent with this, we also observed that siRNA-mediated depletion of PRMT3 significantly increased the expression of all the 13 tested RA-responsive genes in HEK293 cells treated with RALD (Fig. 6). In the absence of RALD, there were no changes or mild increases in gene expression indicating the effect of PRMT3 depletion on the increase of RA-responsive

quantified luciferase activity of retinoic acid response element (RARE)—luciferase reporter expression construct in HEK293 cells in which PRMT3 levels were perturbed and treated with RALD. The construct was designed such that the expression of the luciferase reporter gene was under the control of the promoter containing the RARE element. To ensure that the regulation of luciferase expression is RARE-dependent, we quantified the luciferase activity in HEK293 cells with or without the treatment of RALD. We found that the treatment of RALD increased the luciferase activity by ~23-fold (Supplementary Fig. 7), indicating that an increase in RALD could lead to an increase in RA and

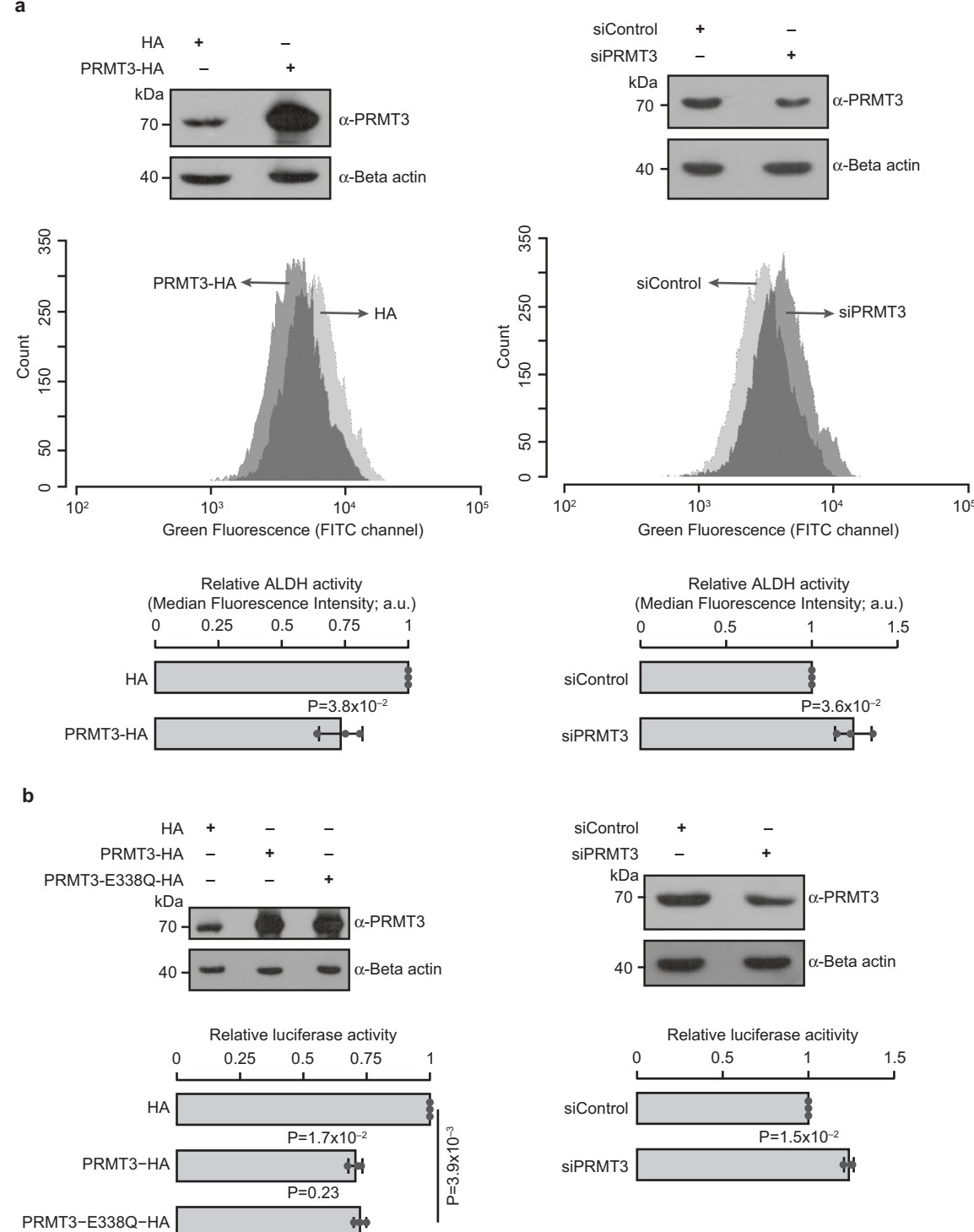

gene expression is RALD-dependent. The finding that PRMT3 perturbation affects the expression of RA-responsive genes in a RALD-dependent manner, suggests that PRMT3 regulates the expression of RA-responsive genes by inhibiting the enzymatic activity of ALDH1A1. Such regulation has far-reaching implications, as it influences diverse biological processes.

**PRMT3 could modulate global effects on retinoic acid-mediated gene-expression regulation.** Based on the above findings, we hypothesized that RA targets will be downregulated in

conditions in which PRMT3 levels are high, regardless of higher ALDH1A1 levels. This is based on the assumption that the high levels of PRMT3 would ensure adequate molecules to bind and inhibit ALDH1A1 activity, with the functional outcome being downregulation of RA targets. On the contrary, low levels of PRMT3 means that ALDH1A1 would be in an unbound state, facilitating transcription of RA target genes, reflected by positive fold changes in the expression levels of RA target genes. To test this hypothesis, we investigated published high-throughput gene-expression profiling datasets obtained in diverse conditions from Expression Atlas[46]. We identified two datasets that matched our

**Fig. 5 PRMT3 inhibits the ALDH1A1 activity in vivo. a** ALDEFLUOR assay reveals that PRMT3 inhibits the ALDH1A1 activity in vivo. HEK293 cells were transfected with HA vector or HA-tagged PRMT3 construct or control siRNA or PRMT3 siRNA. The efficiency of PRMT3 overexpression (upper left blots) and knockdown (upper right blots) was quantified by immunoblotting with PRMT3 antibody. ALDEFLUOR assay was performed in the PRMT3 perturbed cells by incubating with ALDEFLUOR reagent. Representative graphs show the ALDEFLUOR fluorescence intensities of the cells which were transfected with HA vector or HA-tagged PRMT3 construct (middle left panel) or control siRNA or PRMT3 siRNA (middle right panel). The lower panel graphs represent the relative mean fluorescence intensities of three biologically independent ALDEFLUOR experiments. Error bars indicate the standard deviations of the mean. The statistical significance was assessed by a two-tailed $t$ test. **b** PRMT3 reduces the expression of the RARE-luciferase reporter gene. HEK293 cells were co-transfected with HA vector or HA-tagged PRMT3 construct or HA-tagged PRMT3-E338Q construct or control siRNA or PRMT3 siRNA with pGL3-RARE-luciferase and pRL-Renilla-luciferase reporter vector and treated with 2.5 μM of all-trans-retinal (RALD). The efficiency of PRMT3 or PRMT3-E338Q overexpression (upper left blots) and knockdown (upper right blots) was quantified by immunoblotting with PRMT3 antibody. Luciferase reporter activities were quantified in these cells. The firefly luciferase activities were normalized to Renilla luciferase activities and presented relative to the control sample in the graphs (lower panel). The values in the graphs represent the mean of three biologically independent experiments, with error bars representing standard deviations. The statistical significance was assessed by a two-tailed $t$ test (Supplementary Fig. 10 and Supplementary Data 1).

selection criteria; gene-expression changes in (i) young cells compared to spontaneously immortal cells in which ALDH1A1 was upregulated, while PRMT3 was downregulated (referred hereafter as $AL1^{High}PR3^{Low}$) and (ii) lung cancer compared to normal lung cells in which both ALDH1A1 and PRMT3 were upregulated ($AL1^{High}PR3^{High}$, Fig. 7a). We obtained the list of known RA targets from literature[43]. We analyzed the expression levels of positively regulated targets of RA that were found in both the conditions. Since there were very few known downregulated RA targets ($n = 5$), we could not consider this class for further analysis.

We found that in $AL1^{High}PR3^{Low}$ condition, RA targets showed a significantly higher distribution of positive fold changes, while $AL1^{High}PR3^{High}$ displayed a significantly higher distribution of negative fold changes (Median $log_2$ fold changes of 2.5 compared to −2.6; Fig. 7b–d). Importantly, there was a significant enrichment of upregulated targets in $AL1^{High}PR3^{Low}$ (70%) while $AL1^{High}PR3^{High}$ was enriched for downregulated targets (73% of genes; Fig. 7c). The observed effects on the targets in the two conditions cannot be attributed to the difference in the fold change in upregulation of ALDH1A1 ($log_2$ fold change of 3.8 in $AL1^{High}PR3^{Low}$ versus 1.4 in $AL1^{High}PR3^{High}$; Fig. 7a), as such a difference would be expected to show changes in magnitude but not in the directionality of effect on the targets.

RA targets involved in diverse biological processes are negatively regulated by PRMT3. For instance, the well-known RA-responsive genes *RAI3*, *TIG1*, and *RARR2*, whose expression is often dysregulated in several cancers[47–56], are all down-regulated in $AL1^{High}PR3^{High}$ (Fig. 7d). These investigations provide an independent confirmation that PRMT3 inhibits ALDH1A1 dependent RA-mediated global gene-expression regulation. Furthermore, they also establish that such an inverse relationship in the abundance of PRMT3 and ALDH1A1 has physiological and/or pathological relevance in the regulation of expression of multiple targets. Taken together, our experimental and computational investigations clearly indicate that PRMT3 negatively regulates ALDH1A1-mediated retinoic signaling with effects at the transcriptome level, which could impact diverse cellular processes (Fig. 8).

## Discussion
Our findings have far-reaching implications on the physiological roles of PRMT3 on diverse cellular processes regulated by ALDH1A1, such as development, differentiation, adipogenesis, gluconeogenesis, initiation of meiosis, and synthesis of γ-aminobutyric acid (GABA) in dopamine neurons[25–29,57]. For instance, PRMT3-mediated inhibition of ALDH1A1 activity might lead to the accumulation of retinaldehyde, which in turn might inhibit adipogenesis[26]. Importantly, recent findings suggest that the treatment of palmitate promotes lipogenesis in the liver

and PRMT3 has been shown to translocate to the nucleus upon palmitic acid treatment[11]. In light of our results, we propose that the translocation of PRMT3 to the nucleus contributes to the removal of inhibitory PRMT3 from the cytosol, where the ALDH1A1 acts, thereby promoting lipogenesis. Furthermore, our investigations throw light on the possibility that different PRMTs and ALDHs might also interact. This necessitates the complete deconvolution of the PRMT–ALDH nexus and its impact on gene/protein regulation and function.

From a pathological standpoint, high levels of ALDH1A1 are often observed in various cancers. However, there is a contradiction as to whether elevated levels of ALDH1A1 is associated with poor or favorable clinical outcomes in different cancer types[30,31]. Our results show that PRMT3 regulates the activity and not the levels of ALDH1A1. This implies that ALDH1A1 activity might not be a linear function of its abundance. This is well reflected in the lung cancer data analyzed here (Fig. 6) wherein there is an upregulation of ALDH1A1; however, with a significant downregulation of RA-responsive genes, possibly owing to the elevated levels of PRMT3. This could explain the discrepancy in the correlation between ALDH1A1 levels and clinical outcomes in cancer. Based on our findings, we posit that ALDH1A1 activity, rather than the levels should be considered for cancer prognosis. Further studies providing mechanistic insights into the functional roles of PRMT3 are required to explore the therapeutic potential of PRMT3 in cancers and other diseases.

In conclusion, our study shows that PRMT3 affects diverse biological processes not only by globally regulating protein function by introducing methylation marks but also by regulating gene expression through protein–protein interactions.

## Methods
**Cloning, expression, and purification**. The sequence encoding full-length human PRMT3 (PR-FL; NM_005788.3) and the full-length human ALDH1A1 (AL-FL; NM_000689.3) were amplified from cDNA prepared from HEK293 cells. These were cloned into pGBKT7 vector (Clontech) in frame with Gal4 DNA-binding domain using XmaI and SalI sites to generate pGBKT7-PRMT3 construct and pGADT7 vector in frame with GAL4 activation domain to generate pGADT7-ALDH1A1 construct using EcoRI and XhoI sites respectively. PR-FL, N-terminal region of PRMT3 (PR-NTR; amino acids 1–186) and C-terminal region of PRMT3 (PR-CTR; amino acids 186–531) were subcloned in pEGFP-C1 vector (Clontech) using XhoI and BamHI sites to generate pEGFP-PR-FL, pEGFP-PR-NTR, and pEGFP-PR-CTR constructs, respectively. PR-FL, PR-NTR, and PR-CTR were also subcloned in a bacterial expression cassette pGEX-6P2 vector (GE Healthcare) using BamHI and XhoI sites to generate pGEX-PR-FL, pGEX-PR-NTR, and pGEX-PR-CTR constructs, respectively. PR-CTR was also subcloned in bacterial expression cassette pET28a vector (Novagen) using BamHI and XhoI sites to generate pET28-PR3-CD construct.

The oligo encoding the HA tag was introduced into the pCDNA4/myc-HisA vector (Invitrogen) using XhoI and ApaI sites to generate the pCDNA4-HA vector. PR-FL and AL-FL were also subcloned in pCDNA4-HA vector using BamHI and XhoI sites in frame with the C-terminal HA tag to generate pCDNA4-HA-PR-FL and pCDNA4-HA-AL-FL constructs. AL-FL was subcloned into pcDNA4/myc-His

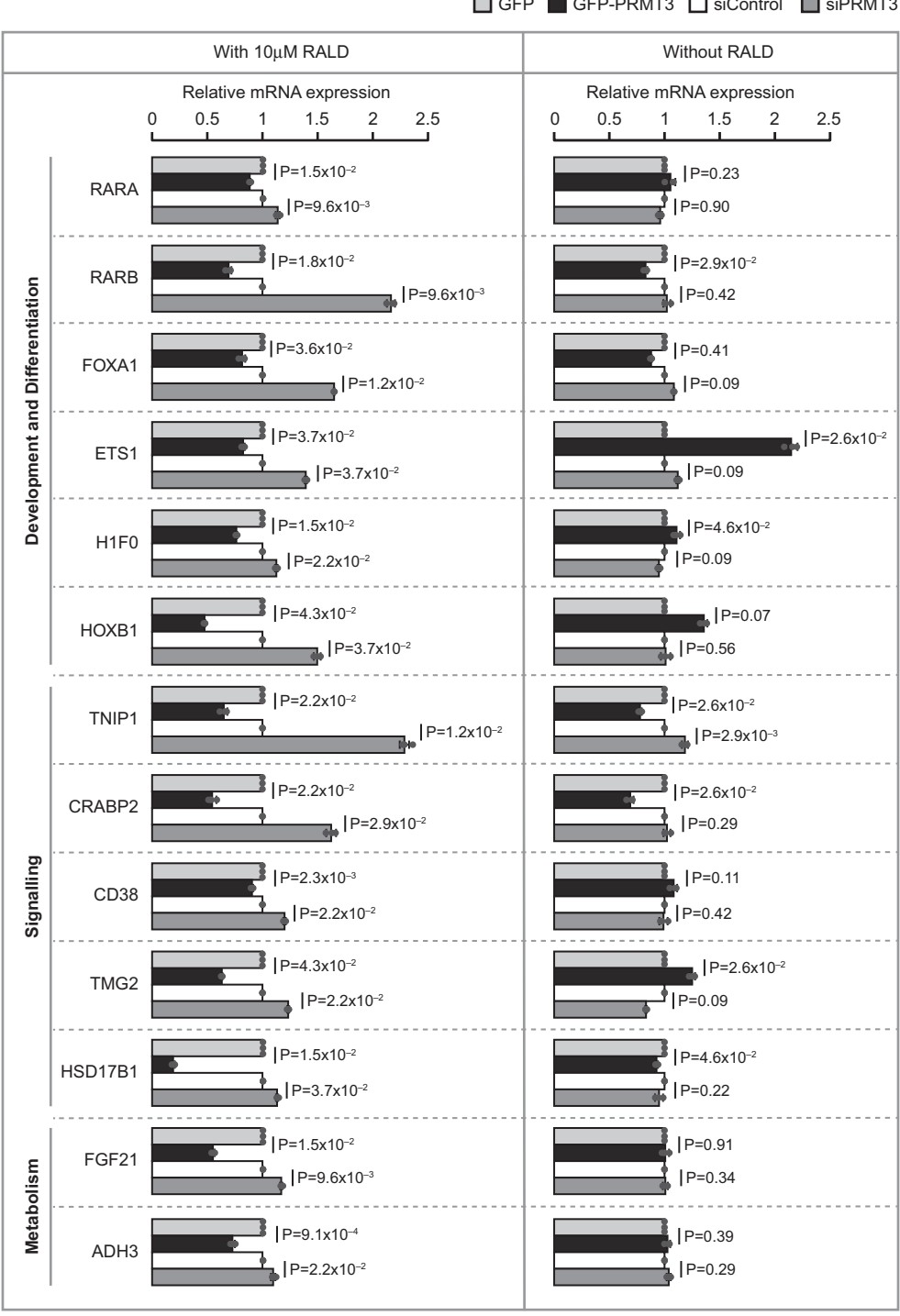

**Fig. 6 PRMT3 negatively regulates the expression of RA-responsive genes.** HEK293 cells were transfected with GFP vector or GFP-tagged PRMT3 construct or control siRNA or PRMT3 siRNA and treated with or without 10 μM of all-trans-retinal (RALD). The graph in the figure presents the mRNA levels of the indicated RA-responsive genes as relative expression quantified by quantitative RT-PCR. The values in the graphs represent the mean of three biologically independent experiments, with error bars representing standard deviations. The statistical significance was assessed by a two-tailed $t$ test, and the false discovery rate (FDR) values were obtained upon correcting the $P$ values for multiple testing using Benjamini–Hochberg method (Supplementary Data 1).

A vector (Invitrogen) using EcoRI and XhoI sites in frame with the C-terminal Myc/His tag to generate the pCDNA4-myc-AL-FL construct. AL-FL, N-terminal domain of ALDH1A1 (AL-NTR) (amino acids 1–335), and C-terminal domain of ALDH1A1 (AL-CTR) (amino acids 335–501) were subcloned in pGEX-6P2 vector using EcoRI and XhoI sites to generate the pGEX-AL-FL, pGEX-AL-NTR, and pGEX-AL-CTR constructs. AL-FL was also subcloned in pEGFP-C1, pET28a vector, and pDsRed-N1 monomer vectors (Clontech) using XhoI and EcoRI sites, EcoRI and XhoI sites, and KpnI and XhoI sites, respectively to generate pEGFP-

AL-FL, pET28a-AL-FL, and pDsRed-AL-FL constructs. E338Q mutation in pEGFP-PR-FL and pCDNA4-HA-PR-FL constructs and H464A, N465A, R466A, and V468A mutations in pGEX-PR-FL were introduced by using the site-directed mutagenesis method, as previously described[58]. The full-length *PRMT2, PRMT5, PRMT6,* and *PRMT7* were cloned in pEGFP-C1 vector to generate pEGFP-PRMT2 or pEGFP-PRMT5 or pEGFP-PRMT6 or pEGFP-PRMT7 constructs, respectively. The full-length *ALDH1A2, ALDH1A3, ALDH3A1,* and *ALDH2* were cloned in pCDNA4-HA vector to generate pCDNA4-HA-ALDH1A2 or pCDNA4-HA-

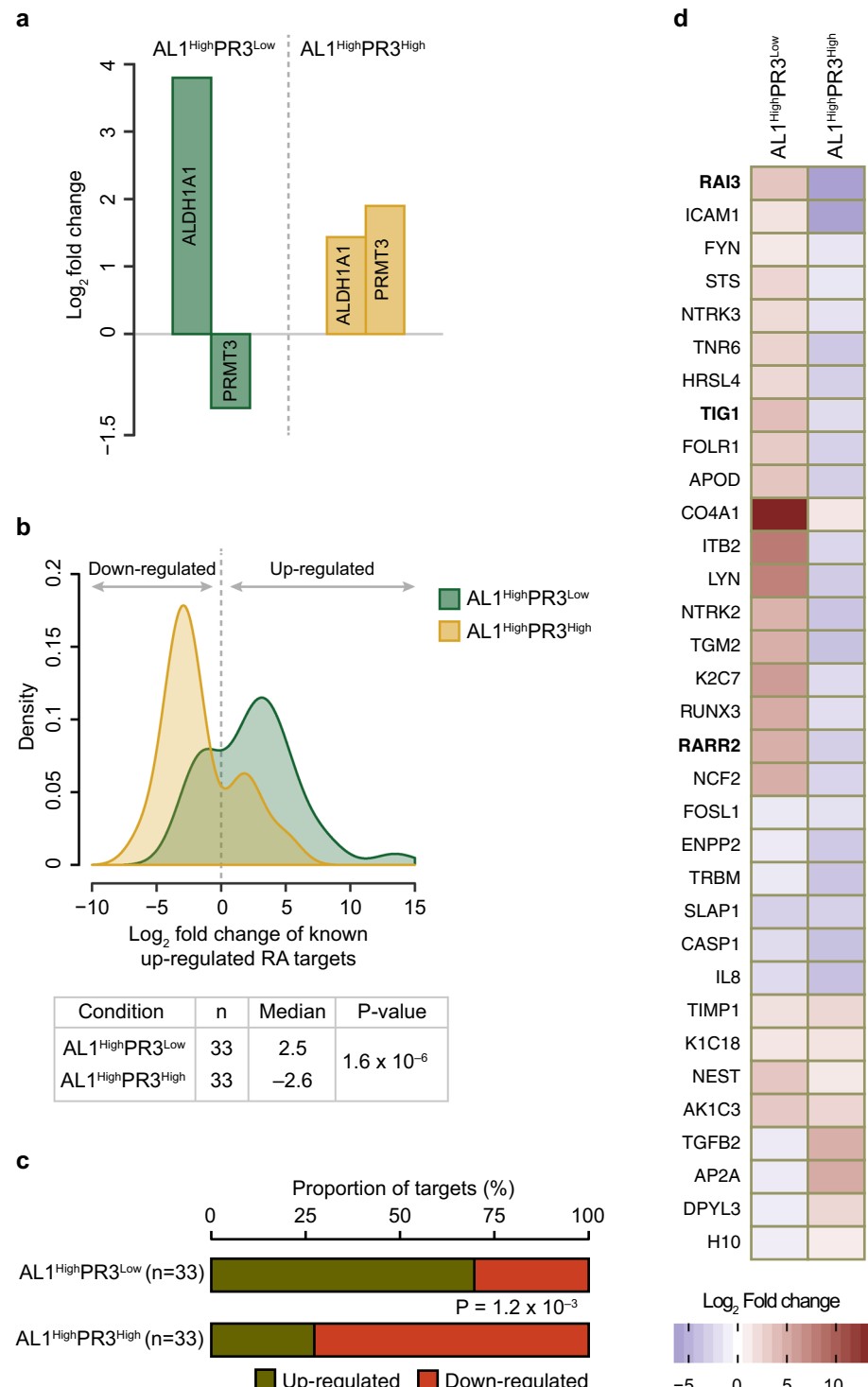

**Fig. 7 Transcriptome-wide analysis across different conditions provides correlation of the levels of PRMT3 and regulation of retinoic acid-responsive genes by ALDH1A1. a** By analyzing different gene-expression profiling studies, we identified two conditions—(i) gene-expression changes in young cells compared to spontaneously immortal cells, in which ALDH1A1 was significantly upregulated and PRMT3 was downregulated (AL1$^{High}$PR3$^{Low}$), and (ii) transcriptional changes in lung cancer compared to normal lung cells, in which both ALDH1A1 and PRMT3 were significantly upregulated (AL1$^{High}$PR3$^{High}$). **b** Distribution of fold changes (log$_2$) in the two conditions of known targets with literature evidence for upregulation. The table under the distribution plot provides the number (*n*) of targets, the median fold change in each condition, and the *P* value signifying differences in the distribution of fold changes in the two conditions. Statistical significance was assessed using the non-parametric Wilcoxon rank-sum test. **c** Distribution of upregulated and downregulated RA targets in AL1$^{High}$PR3$^{Low}$ and AL1$^{High}$PR3$^{High}$ conditions. Statistical significance (depicted by *P* value) was assessed using Fisher's exact test. **d** Heatmap showing the differences in fold changes between the two conditions for targets known to be upregulated by RA. Well-known RA-responsive genes are highlighted in bold text.

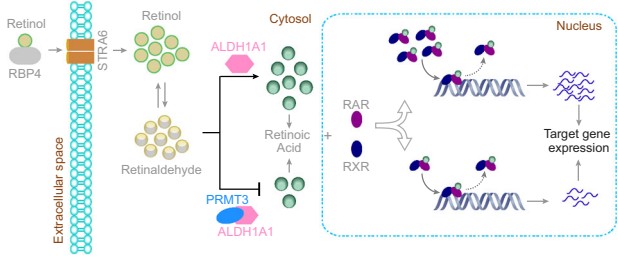

**Fig. 8 Schema illustrating the functional significance of PRMT3–ALDH1A1 interaction.** PRMT3 interacts with ALDH1A1 and inhibits the enzymatic activity of ALDH1A1. This interaction negatively regulates retinoic acid signaling-mediated gene expression.

ALDH1A3 or pCDNA4-HA-ALDH3A1 or pCDNA4-HA-ALDH2 constructs, respectively.

The bacterial expression and purification of GST-tagged PR-FL, PR-NTR, PR-CTR, AL-FL, AL-NTR, and AL-CTR were performed as described previously[59], and bacterial expression and purification of His-tagged AL-FL and His-PR3-CD were performed as described previously[60]. The purified proteins were loaded in 12% SDS-PAGE to assess the purity of the proteins.

**Yeast two-hybrid assay.** The yeast two-hybrid screening was performed using Matchmaker Gold Yeast Two-Hybrid System (Clontech) using pGBKT7-PR-FL construct and human cDNA library (Mate & Plate Library—Universal Human (Normalized); Clontech) according to the manufacturers' instructions. To confirm the interaction of full-length ALDH1A1 with PRMT3, pGBKT7-PRMT3, and pGADT7-ALDH1A1; pGBKT7 and pGADT7-ALDH1A1, and pGBKT7-PRMT3 and pGADT7 constructs were co-transformed in *S. cerevisiae* MM Gold strain and plated in a selection medium containing Aureobasidin-A (Ab-A) and X-α-Gal but lacking tryptophan and leucine or tryptophan, leucine, histine, and adenine to assess the expression of two or all the four reporter genes.

**Cell culture and transfection.** HEK293 cells were purchased from NCCS, Pune, India and cultured in DMEM (Himedia) supplemented with 10% FBS (Himedia), sodium pyruvate, and L-glutamine–penicillin–streptomycin solution (Himedia) at 37 °C in 5% $CO_2$, and the transfections were performed using the standard calcium phosphate precipitation method. The control siRNA (siRNA negative control, Eurogentec, SR-CL000-005) and the PRMT3 siRNA (5′-GGA UGA GGA UGG UGU UUA U-3′) were transfected using Lipofectamine 2000.

**Co-immunoprecipitation, immunoprecipitation, and co-localization experiments.** The co-immunoprecipitation (Co-IP), immunoprecipitation, and co-localization experiments were performed, as described previously[60]. For the forward Co-IP experiments, Myc-tagged ALDH1A1 (pCDNA4-myc-AL-FL construct) was co-transfected with either pEGFP-C1 vector or GFP-tagged PRMT3 (pEGFP-PR-FL construct) in HEK293 cells. HA-tagged PRMT3 (pCDNA4-HA-PR-FL construct) was co-transfected with either pEGFP-C1 vector or GFP-tagged ALDH1A1 (pEGFP-AL-FL construct) in HEK293 cells for the reverse Co-IP experiments. After 48 h of transfection, the cells were lysed in a lysis buffer (10 mM Tris, pH: 7.5, 150 mM NaCl, 0.5 mM EDTA, 0.5% NP-40, and Protease Inhibitor Cocktail (Roche)), and the cell lysates were diluted with dilution buffer (10 mM Tris, pH: 7.5, 150 mM NaCl, and 0.5 mM EDTA) to reduce the NP-40 concentration to 0.1%. The cell lysates were incubated with GFP Trap A beads (Chromotek) at 4 °C for 10 h with rotation in a roller. After the incubation, the beads were washed thrice in a wash buffer (10 mM Tris, pH: 7.5, 150 mM NaCl, 0.5 mM EDTA, and 0.1% NP-40) and boiled with 2× LAP for 5 min for elution. The eluted proteins were resolved in 12% SDS-PAGE and transferred to the nitrocellulose membrane. The membrane was probed with Myc antibody (Santa Cruz Biotechnology; clone 9E10; 1 in 250 dilutions) in case of forward Co-IP or anti-HA high-affinity antibody (Roche; Cat. No. 11867423001; 1 in 1000 dilution) in case of reverse Co-IP.

For the immunoprecipitation experiments, the HEK293 cell lysates were incubated with rabbit IgG (Cell Signaling Technology; Cat. No. 2729) or ALDH1A1 antibody (Proteintech, Cat. No. 22109-1-AP) or at 4 °C for 1 h. After the incubation, Protein A dynabeads were added and incubated further for 10 h at 4 °C with rotation in a roller. After the incubation period, the beads were washed extensively with wash buffer (10 mM Tris, pH: 7.5, 150 mM NaCl, 0.5 mM EDTA, and 0.2% NP-40), and the bound fractions were eluted by boiling with 2× LAP. The eluted fractions were immunoblotted with PRMT3 monoclonal antibody (Abcam; Cat. No. ab191562; 1 in 3000 dilutions).

For the co-localization experiments, the HEK293 cells were seeded in coverslips and co-transfected with pEGFP-PR-FL and pDsRed-AL-FL constructs. After 48 h of transfection, the cells were fixed with 4% formaldehyde for 10 min at room temperature, treated with DAPI (Sigma), and embedded with Mowiol (Sigma). Confocal images were taken using a Zeiss LSM 510 Meta instrument (software version 3.0) and ×63 oil immersion objective.

**GST pull-down and Ni-NTA pull-down assays.** The GST and Ni-NTA pull-down experiments were performed as described previously[60,61]. About 15 µg of GST or GST-tagged PRMT3 or PRMT3-H464A or PRMT3-N465A or PRMT3-R466A or PRMT3-V468A recombinant proteins were coupled with 25 µl of Glutathione Sepharose 4 Fast Flow resin (GE Healthcare) in the ice-cold interaction buffer (20 mM HEPES, pH: 7.5, 150 mM KCl, 0.2 mM DTT, 1 mM EDTA, and 10% glycerol). The protein-coupled beads were blocked at 4 °C for 2 h with a blocking buffer (interaction buffer containing 5% BSA). After the blocking step, the beads were incubated with the 15 µg of recombinant His-tagged AL-FL protein and with or without 200 µM of SAM or SAH at 4 °C for 10 h with rotation in the roller. After the incubation, the beads were washed thrice with wash buffer (10 mM Tris, pH: 7.5, 150 mM NaCl, 0.5 mM EDTA, and 0.1% NP-40) and boiled with 2× LAP for 5 min for elution. The bound fraction was loaded in 12% SDS-PAGE, and the resolved proteins were transferred to the nitrocellulose membrane and probed with anti-His antibody (GE Healthcare; 27-4710-01; 1 in 3000 dilutions).

For the Ni-NTA pull-down assay, 15 µg of His-tagged AL-FL was coupled with 25 µl of Ni-NTA resin (Clontech) in the interaction buffer (20 mM HEPES, (pH: 7.5), 150 mM KCl, 0.2 mM DTT, 1 mM EDTA, and 10% glycerol). For the control pulldown, 25 µl of empty Ni-NTA resin was used. The beads were blocked at 4 °C for 2 h with a blocking buffer (interaction buffer containing 5% BSA). After the blocking step, the beads were incubated with the 15 µg of GST-tagged PR-FL protein at 4 °C for 10 h with rotation in the roller. After incubation, the beads were washed and processed as described in the GST pull-down assay. The bound fractions were analyzed by immunoblotting using an anti-GST antibody (GE Healthcare; Cat. No. 27-4577-01; 1 in 2000 dilution).

For identifying the domains involved in the interaction, 15 µg of GST or the GST-tagged AL-FL or AL-NTR or AL-CTR recombinant proteins were coupled with 25 µl of Glutathione Sepharose 4 Fast flow resin (GE Healthcare) in the ice-cold interaction buffer (20 mM HEPES, pH: 7.5, 150 mM KCl, 0.2 mM DTT, 1 mM EDTA, and 10% glycerol). The protein-coupled beads were blocked at 4 °C for 2 h with a blocking buffer (Interaction buffer containing 5% BSA). After the blocking step, the beads were incubated with the cell lysates prepared from the cells, which were transfected with pEGFP-PRMT3-FL or pEGFP-PR-NTR or pEGFP-PR-CTR constructs at 4 °C for 10 h with rotation in the roller. The cell lysates were prepared and diluted to 0.1% NP-40, as mentioned above. After the incubation, the beads were washed thrice with wash buffer (10 mM Tris, pH: 7.5, 150 mM NaCl, 0.5 mM EDTA, and 0.1% NP-40) and boiled with 2× LAP for 5 min for elution. The bound fraction was loaded in 12% SDS-PAGE, and the resolved proteins were transferred to the nitrocellulose membrane and probed with an anti-GFP antibody (Clontech; Cat. No. 632381; 1 in 3000 dilutions).

**GST pull-down assays to study the interaction of ALDH1A1 with different PRMT members and that of PRMT3 with different ALDH members.** To study the interaction of PRMT3 with other ALDH members, 15 µg of GST or the GST-tagged PRMT3 recombinant proteins were coupled with 25 µl of Glutathione Sepharose 4 Fast Flow resin (GE Healthcare) in the ice-cold interaction buffer (20 mM HEPES, pH: 7.5, 150 mM KCl, 0.2 mM DTT, 1 mM EDTA, and 10% glycerol). The protein-coupled beads were blocked at 4 °C for 2 h with a blocking buffer (interaction buffer containing 5% BSA). After the blocking step, the beads were incubated with the cell lysates prepared from the cells, which were transfected with pCDNA4-HA-ALDH1A1 or pCDNA4-HA-ALDH1A2 or pCDNA4-HA-ALDH1A3 or pCDNA4-HA-ALDH3A1 or pCDNA4-HA-ALDH2 constructs at 4 °C for 10 h with rotation in the roller. The cell lysates were prepared and diluted to 0.1% NP-40, as mentioned above. After the incubation, the beads were washed thrice with wash buffer (10 mM Tris, pH: 7.5, 150 mM NaCl, 0.5 mM EDTA, and 0.1% NP-40) and boiled with 2× LAP for 5 min for elution. The bound fraction was loaded in 12% SDS-PAGE and the resolved proteins were transferred to the nitrocellulose membrane and probed with anti-HA antibody (Roche; Cat. No. 11867423001; 1 in 1000 dilution).

To study the interaction of ALDH1A1 with other PRMT members, 15 µg of GST or the GST-tagged ALDH1A1 recombinant proteins were coupled with 25 µl of Glutathione Sepharose 4 Fast Flow resin (GE Healthcare) in the ice-cold interaction buffer (20 mM HEPES, pH: 7.5, 150 mM KCl, 0.2 mM DTT, 1 mM EDTA, and 10% glycerol). The protein-coupled beads were blocked at 4 °C for 2 h with a blocking buffer (Interaction buffer containing 5% BSA). After the blocking step, the beads were incubated with the cell lysates prepared from the cells, which were transfected with pEGFP-PRMT3 or pEGFP-PRMT2 or pEGFP-PRMT5 or pEGFP-PRMT6 or pEGFP-PRMT7 constructs at 4 °C for 10 h with rotation in the roller. The cell lysates were prepared and diluted to 0.1% NP-40, as mentioned above. After the incubation, the beads were washed thrice with wash buffer (10 mM Tris, pH: 7.5, 150 mM NaCl, 0.5 mM EDTA, and 0.1% NP-40) and boiled with 2× LAP for 5 min for elution. The bound fraction was loaded in 12% SDS-PAGE and the resolved proteins were transferred to the nitrocellulose membrane and probed with an anti-GFP antibody (Clontech; Cat. No. 632381; 1 in 3000 dilutions).

**Ortholog detection and conservation analysis**. We obtained sequence orthologs of human PRMT3 ($n = 196$) and ALDH1A1 ($n = 4372$) from OMA orthology database[62]. We ensured that each organism was represented by only one sequence by disregarding orthologs from multiple strains from the same species. Since aldehyde dehydrogenase activity is a fundamental metabolic function, there were multiple orthologs and paralogs for ALDH1A1 across diverse organisms. To curate one-to-one orthologs, we obtained pair-wise sequence identity estimates for all orthologs of human ALDH1A1 by generating sequence alignments using MAFFT[63]. In each organism, the ortholog with the highest sequence identity with the human ALDH1A1, greater than that with any other human paralog was assigned as the one-to-one ortholog. Using such stringent selection criteria, we identified 94 one-to-one orthologs of human ALDH1A1, spanning across bacteria, archaea, and eukaryotes. Similarly, for human PRMT3 we considered only one-to-one orthologs and disregarded other types of orthologs such as one-to-many or many-to-one orthologs and paralogs arising from gene-duplication event(s) in any species. PRMT3 orthologs were found only in eukaryotes. Then, we classified organisms into those with both ALDH1A1 and PRMT3 (ALDH1A1+PRMT3+) and those in which ALDH1A1 was present and PRMT3 could not be detected (ALDH1A1+PRMT3−). We then generated multiple sequence alignments for ALDH1A1 orthologs belonging to the two sets using MAFFT. Using the alignment, we then estimated Jenson–Shanon Divergence (JSD) for residues in the C-terminal region of human ALDH1A1, using the protein residue prediction server[64]. We derived the background amino acid frequencies for sites subject to no evolutionary pressure using the BLOSUM62 matrix. For computing JSD, we used the default settings of window size of 3 and opted for sequence weighting that rewards sequences that are "surprising". The JSD score ranges between 0 and 1, with the higher scores indicating positions under evolutionary pressure and hence their functional importance.

**Molecular docking studies of PRMT3 and ALDH1A1**. To identify the hotspot regions and key residues involved in human PRMT3 and human ALDH1A1 interaction, crystal structures of human PRMT3 (PDB ID: 2FYT) and human ALDH1A1 (Retinal dehydrogenase 1; PDB ID: 5L2M) are docked globally using HAWKDOCK server ATTRACT algorithm[65–67]. A total of 100 protein–protein interaction models are predicted, and the best-docked conformation is selected based on the HawkRank re-scoring algorithm[68] and visual structural inspection. Further, binding free energies for the important PPI residues are estimated by in-built MM/GBSA free energy decomposition analysis algorithm[69–71].

**ALDH1A1 enzymatic activity assay**. ALDH1A1 activity assay was set up in a reaction buffer (20 mM HEPES; pH: 7.4, 120 mM NaCl) containing 0.5 µM of GST-tagged ALDH1A1 full-length enzyme, 485 µM of propionaldehyde (Sigma), supplemented with and without 0.5 µM of full-length GST-tagged PRMT3 (GST-PRMT3-FL) or N-terminal domain of PRMT3 (GST-PR-NTR) or C-terminal domain of PRMT3 (GST-PR-CTR). To test whether the effects of PRMT3 on ALDH1A1 is dependent on PRMT3 concentration, we set up the concentration-dependent inhibition assay without or with 0.23 µM or 0.45 µM or 0.68 µM or 0.9 µM or 1.13 µM of GST-PRMT3-FL protein. The reaction mixture was incubated at 4 °C for 45 min to allow the binding of the PRMT3 protein with ALDH1A1, and then the reaction mixture is incubated at room temperature for 15 min. The reaction was initiated by adding the 125 µM of NAD+ cofactor (NEB). For the control reaction, NAD+ was excluded. Accumulation of NADH as a result of enzymatic activity was measured spectrophotometrically at 340 nm at the kinetic time intervals of 2 min for 60 min, and the concentration was extrapolated from a standard curve of NADH. The mean of NADH concentration for three independent experiments was plotted against time. The rate of the reaction represents the mean of slopes of the three independent experiments and the error bar indicates the standard deviation of the mean.

**Methylation assay**. Methylation assay was performed in PBS (pH: 7.4) by incubating 2.5 µg of GST or 2.5 µg of GST-tagged ALDH1A1 or 1 µg of GST-GAR proteins with or without 6 µg of His-tagged catalytic domain of PRMT3 (His-PRMT3-CD) in the presence or absence of 0.76 µM radiolabeled S-adenosyl-L-[methyl-$^3$H] methionine (SAM). The reaction mixture was incubated at 37 °C for 2 h and then separated in 12% SDS-PAGE. The gel was incubated with fluorographic reagent (GE Healthcare, NAMP100); dried and exposed to X-ray film for 14 days at −80 °C.

**Aldefluor assay**. HEK293 cells were transfected with pCDNA4-HA vector or pCDNA4-HA-PR-FL construct or control siRNA or PRMT3 siRNA. After 48 h of transfection, the cells were treated with the Aldeflour reagent (ALDEFLUOR Kit, Stemcell Technologies) as per the manufacturers' instructions. The fluorescence signal of the cells was quantified in FACS AriaIII flow cytometer (BD Biosciences, USA) in the green fluorescent channel using the software FACS Diva version 8.

**Luciferase assay**. HEK293 cells were co-transfected with pCDNA4-HA vector or pCDNA4-HA-PR-FL construct or pCDNA4-HA-PR-FL-E338Q construct or control siRNA or PRMT3 siRNA with pGL3-RARE-luciferase[72] (Addgene, Plasmid No. 13458) and pRL-Renilla luciferase reporter vector (pRL-CMV vector,

Promega) mixture in a ratio of 50:1. After 24 h of transfection, the cells were treated with or without 2.5 µM of all-trans-retinal (RALD). After 48 h of transfection, the cells were lysed, and luciferase activity assay was performed by using a dual-luciferase reporter assay kit (Promega, Cat. No. E1910) as per the manufacturer's instructions.

**Real-time PCR**. For the Quantitative Real-Time PCR (qRT-PCR) assays, HEK293 cells were transfected with either pEGFP-C1 vector or pEGFP-PR-FL construct or pEGFP-PR-FL-E338Q construct or control siRNA or PRMT3 siRNA. About 24 h post transfection, the cells were treated with or without 10 µM of RALD. After 48 h upon transfection, the cells were washed with PBS, and the total RNA was isolated using Trizol reagent (Life Technologies) according to the manufacturers' instructions. The total RNA was reverse transcribed, and the qRT-PCR analysis was performed as described previously[73]. The primers that were used in the qRT-PCR experiment are listed in Supplementary Table 3.

**Transcriptome-wide analysis**. We obtained published datasets profiling differential expression of ALDH1A1 and PRMT3 from Expression Atlas[46]. ALDH1A1 and PRMT3 were significantly differentially expressed in 133 and 54 conditions, respectively. We selected conditions in which (i) ALDH1A1 was upregulated but PRMT3 was downregulated (AL1HighPR3Low) or (ii) both ALDH1A1 and PRMT3 were upregulated (AL1HighPR3High). Gene-expression changes in young cells (replicative) compared to spontaneously immortal (induced senescence) cells[74] (Expression Atlas expression accession: E-GEOD-60340) showed significant upregulation of ALDH1A1 and downregulation of PRMT3 (AL1HighPR3Low). Transcriptional changes in lung cancer compared to normal lung cells from freshly frozen human tissues[75] (E-GEOD-19249), showed significant upregulation of both ALDH1A1 and PRMT3 were significantly upregulated (AL1HighPR3High). We obtained known retinoic acid (RA) regulatory targets from literature, which also documents whether a target is upregulated or downregulated[43]. We drew comparisons of the distribution of fold changes of targets that have been previously shown to be upregulated by RA (known upregulated targets) that were present in both conditions. Continuous distributions (Fig. 7b) were tested for statistical significance using the non-parametric Wilcoxon rank-sum test, and the distribution of discrete variables (Fig. 7c) was compared using Fisher's exact test.

**Statistics and reproducibility**. The details of statistical tests and the number of replicates are provided in the figure legends and in "Methods". The data were analyzed using R and Microsoft Excel. The values in the graphs represent the mean of at least three independent experiments, with error bars representing standard deviation. $P$ values and FDR values < 0.05 were considered to be statistically significant.

**Reporting summary**. Further information on research design is available in the Nature Research Reporting Summary linked to this article.

## Data availability
All the data are available from the corresponding author (Arunkumar Dhayalan) on request. There are no restrictions on the data availability. The accession of the human PRMT3 structure used for molecular docking is PDB ID: 2FYT and that for human ALDH1A1 is PDB ID: 5L2M. The analyzed transcriptomics data were obtained from Expression Atlas. The expression accession for the dataset on gene-expression changes in young cells (replicative) compared to spontaneously immortal (induced senescence) cells is E-GEOD-60340 and for transcriptional changes in lung cancer compared to normal lung cells from freshly frozen human tissues is E-GEOD-19249. Source Data for graphs are available in Supplementary Data 1.

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

## Acknowledgements
This work was funded by the Innovative Young Biotechnologist Award (Grant No. BT/03/IYBA/2010; A.D. and M.V.), Ramalingaswami Re-entry Fellowship (BT/RLF/Re-entry/33/2018; P.L.C., BT/RLF/Re-entry/05/2018; S.C.), Department of Biotechnology; Science & Engineering Research Board (Grant No. SR/FT/LS-28/2012; A.D.; SRG/2019/001785; S.C.); IISER Tirupati postdoctoral fellowship (R.V.K.); University Grants Commission (UGC-BSR Junior and Senior Research Fellowships to M.I.K.K. and A.M.); Council of Scientific and Industrial Research, Government of India (Junior and Senior Research Fellowships to B.C. and S.A.); RGCB Intramural grant (A.R.) and IISER Tirupati seed grant (S.C.). We thank DST-FIST and UGC-Special Assistant Programme (SAP) funded instrumentation facilities and the BUILDER Instrumentation Facility, Pondicherry University. We thank Dr. Shalmoli Bhattacharyya, PGIMER, Chandigarh for kindly sharing the ALDEFLUOR Reagent for preliminary experiments.

## Author contributions
M.V., M.I.K.K., S.C., and A.D. designed the study. M.V., M.I.K.K., B.C., S.A., A.M., and G.G. performed the experiments. S.C. and R.V.K. performed the bioinformatics analyses. M.V., M.I.K.K., R.V.K., P.L.C., A.R., S.C., and A.D. interpreted the results. M.V., M.I.K.K., R.V.K., S.C., and A.D. wrote the paper with inputs from all the authors. S.C. and A.D. supervised the study.

## Competing interests
The authors declare no competing interests.
