## [Peer Review File · Communications Biology]

Reviewers' comments:

Reviewer #1 (Remarks to the Author):

Verma et al. in this manuscript reported that PRMT3 interacts with ALDH1A1 and inhibits its activity, which leads to negative regulation of the expression of retinoic acid responsive genes. The text is well written and organized. The experiments are carefully designed and mostly contain meaningful controls. The data are solid and clearly presented. In PRMT field, most attentions are on methylase activity-driven biology. This work demonstrates PRMT3 exerts its function on ALDH1A1 by sole protein-protein interactions. The finding is novel and will be an important publication of broad interest to the community.

Many aftermath questions remain to be answered. I list a few here and the authors should consider to discuss some of them in the manuscript.

The structural basis of PRMT3-ALDH1A1 interaction should be determined. Does SAM or SAH binding affect PRMT3-ALDH1A1 interaction? The catalytic domain of PRMTs are highly conserved structurally. Do other PRMTs bind ALDH1A1 as well? Or other PRMTs bind to other ALDH members?

Reviewer #2 (Remarks to the Author):

This manuscript by Verma et al, describes a novel interaction between the protein arginine methyltransferase PRMT3 and the catalyst of retinoic acid, ALDH1A1. The authors use a number of approaches to characterize the interaction between these diverse proteins and understand its relevance to retinoic-acid mediated gene expression. Overall, the manuscript provides sufficient evidence through well executed experiments to support the authors' model. The work may be suitable for publication in Communications Biology, with a few specific comments that should be addressed:

- The authors claim that regulation of ALDH1A1 by PRMT3 is independent of arginine methylation, however the experiments themselves may not rule out a methylation-dependent regulation. For example, on page 6 the authors show that while the C terminal domain of PRMT3 (where the catalytic domain resides) can inhibit ALDH1A1 activity, this is unlikely dependent on catalytic activity, since no SAM was included. However, SAM may be present as a carry-over contaminant from the preparations of the protein fragments. Moreover, while PRMT3 cannot methylate ALDH1A1 in an in vitro methylation assay, this does not exclude the possibility that additional protein co-factors are required for ALDH1A1 methylation by PRMT3 in a cellular context. To show regulation is independent of PRMT3 catalytic activity, experiments should address this directly – either with a catalytically inactive PRMT3 mutant or using a selective inhibitor PRMT3 catalytic activity (SGC707; commercially available).
- The section describing co-evolution of PRMT3 and ALDH1A1 is confusing, and do not contribute to the thesis of the manuscript. While the data does support a eukaryotic form of regulating ALDH1A1 activity, it is not clear these proteins reciprocally affected each other's evolution. This section should be removed (or moved to supplemental).
- On page 8, it is not clear in the following sentence what the fold changes are relative to '...down-regulation of PRMT3 means that ALDH1A1 would be in an unbound state, facilitating transcription of RA target genes, reflected by positive fold changes in the expression levels of RA target genes.' Please clarify.
- The final section is highly speculative without genome-wide functional data. Here, again, use of a catalytically inactive PRMT3 mutant or PRMT3 inhibitor in the presence of RA, followed by microarray (or more extensive quantitative RT-PCR) would better support the authors' model.
- Minor point, but the immunofluorescence in Figure 1F is correlative. While these images demonstrate that both proteins may be compartmentalized within the cytoplasm, the authors present numerous other (much stronger) data to support a PRMT3-ALDH1A1 interaction. These

images may be moved to supplemental.

Point-by-point response to reviewers' comments

Reviewer #1

Comment 1.0: Verma et al. in this manuscript reported that PRMT3 interacts with ALDH1A1 and inhibits its activity, which leads to negative regulation of the expression of retinoic acid responsive genes. The text is well written and organized. The experiments are carefully designed and mostly contain meaningful controls. The data are solid and clearly presented. In PRMT field, most attentions are on methylase activity-driven biology. This work demonstrates PRMT3 exerts its function on ALDH1A1 by sole protein-protein interactions. The finding is novel and will be an important publication of broad interest to the community.

Response 1.0: We thank the reviewer for the positive evaluation of our manuscript and for providing critical inputs that has helped to expand and consolidate our findings.

Comment 1.1: The structural basis of PRMT3-ALDH1A1 interaction should be determined.

Response 1.1: To identify the structural basis of PRMT3-ALDH1A1 interaction, we performed molecular docking and Molecular Mechanics/Generalized Born Surface Area (MM/GBSA) analysis using the crystal structures of human PRMT3 and human ALDH1A1 proteins. We have identified the key hot spot residues in PRMT3 (His464, Asn465, Arg466 and Val468) and their mode of interaction with residues in ALDH1A1. These results are provided as **Fig. 3a and 3b**. We further tested the importance of these residues by mutating each of these residues of PRMT3 located in the interaction interface to alanine. We investigated the interaction of these PRMT3 mutant proteins with ALDH1A1 through GST pull down experiments. We found that these mutations abrogated the PRMT3-ALDH1A1 interaction indicating the importance of these residues, as appropriately identified by our molecular docking studies (**Fig. 3c**).

We have added these data in **Fig. 3** of the revised manuscript and have discussed these results in ‘**The C-terminal residues that lie outside the catalytic active site of PRMT3 facilitate its interaction with ALDH1A1**’ sub-section of the Results section of the revised manuscript (Page 6).

Comment 1.2: Does SAM or SAH binding affect PRMT3-ALDH1A1 interaction?

Response 1.2: We thank the reviewer for this suggestion. We investigated the effect of the SAM or SAH binding to PRMT3 on PRMT3-ALDH1A1 interaction by performing GST pull down assays. We found that addition of SAM or SAH did not affect the PRMT3-ALDH1A1 interaction.

We have added this data in **Fig. 1g** of the revised manuscript and have discussed the findings in ‘**The catalytic domain of PRMT3 interacts with C-terminal region of ALDH1A1**’ sub-section of the Results section of the revised manuscript (Page 5).

Comment 1.3: The catalytic domain of PRMTs are highly conserved structurally. Do other PRMTs bind ALDH1A1 as well? Or other PRMTs bind to other ALDH members?

Response 1.3: We agree with the reviewer that catalytic domains of PRMTs and ALDH members are conserved and it is possible that PRMT3 may interact with other ALDH members and similarly ALDH1A1 may interact with other PRMT members.

To investigate this, we selected representatives from each type of PRMTs (Type I: PRMT2 and PRMT6; Type II: PRMT5 and Type III: PRMT7; **Fig. 2a**). We then tested the interaction of these representative PRMTs with ALDH1A1 through GST pull down assays, using PRMT3 as a positive control. We could not detect any interaction of ALDH1A1 with PRMT2, PRMT5 and PRMT7, while ALDH1A1 showed a feeble interaction with PRMT6, which was very weak compared to the PRMT3-ALDH1A1 interaction (**Fig. 2b**).

For studying the interaction of other ALDHs with PRMT3, we first generated a phylogenetic tree of human ALDH members (**Supplementary Table 1**) by performing multiple sequence alignment. Based on the phylogenetic tree, we selected the other two ALDH1 members (ALDH1A2 and ALDH1A3), ALDH2 which is a closely related member to the ALDH1 clade and ALDH3A1 as a distantly related outgroup member (**Fig. 2c**). We then performed GST pull down experiments to test interaction of ALDH1A2, ALDH1A3, ALDH2 and ALDH3A1 with PRMT3, using ALDH1A1 as a positive control. Interestingly, PRMT3 showed weak interactions with ALDH1A2, ALDH1A3 and ALDH3A1, while we could not detect interaction with ALDH2 (**Fig. 2d**). Notably, these interactions were much weaker compared to PRMT3-ALDH1A1. These findings establish that among the tested PRMT and ALDH members, PRMT3 exhibits the strongest interaction with ALDH1A1.

We have added this data in **Fig. 2** of the revised manuscript and have provided these results in the ‘**PRMT3 exhibits strong binding with ALDH1A1**’ sub-section of the Results section of the revised manuscript (Pages 5 and 6).

Reviewer #2

Comment 2.0: This manuscript by Verma et al, describes a novel interaction between the protein arginine methyltransferase PRMT3 and the catalyst of retinoic acid, ALDH1A1. The authors use a number of approaches to characterize the interaction between these diverse proteins and understand its relevance to retinoic-acid mediated gene expression. Overall, the manuscript provides sufficient evidence through well executed experiments to support the authors' model. The work may be suitable for publication in Communications Biology, with a few specific comments that should be addressed:

Response 2.0: We thank the reviewer for being appreciative of our findings and for the insightful comments, addressing which has strengthened our study further.

Comment 2.1: The authors claim that regulation of ALDH1A1 by PRMT3 is independent of arginine methylation, however the experiments themselves may not rule out a methylation-dependent regulation. For example, on page 6 the authors show that while the C terminal domain of PRMT3 (where the catalytic domain resides) can inhibit ALDH1A1 activity, this is unlikely dependent on catalytic activity, since no SAM was included. However, SAM may be present as a carry-over contaminant from the preparations of the protein fragments. Moreover, while PRMT3 cannot methylate ALDH1A1 in an in vitro methylation assay, this does not exclude the possibility that additional protein co-factors are required for ALDH1A1 methylation by PRMT3 in a cellular context. To show regulation is independent of PRMT3 catalytic activity, experiments should address this directly – either with a catalytically inactive PRMT3 mutant or using a selective inhibitor PRMT3 catalytic activity (SGC707; commercially available).

Response 2.1: We thank the reviewer for raising this point. To address this, we investigated whether the regulatory role of PRMT3 on ALDH1A1 is dependent of PRMT3 catalytic activity or not.

For this, we performed the following new experiments. (i) We quantified the luciferase activity of Retinoic Acid Response Element (RARE) - luciferase reporter expression construct in HEK293 cells in which either wild type PRMT3 or catalytically inactive mutant (E338Q) of PRMT3 was over-expressed. We found that the over-expression of wild type PRMT3 or catalytically inactive mutant PRMT3-E338Q reduced the reporter luciferase activity significantly (~ 30%) compared to the vector control cells (**Fig. 5b**). We did not observe any significant difference between wild type PRMT3 and PRMT3-E338Q mutant on the magnitude of the reduction of luciferase activity. This indicates that PRMT3 negatively regulates RA signaling in a methyltransferase activity independent manner.

We have added this result in **Fig. 5b** of the revised manuscript and have added the text pertaining to this finding in the ‘**PRMT3 negatively regulates ALDH1A1-mediated retinoic acid signaling**’ sub-section of the Results section of the revised manuscript (Pages 8 and 9).

(ii) We also investigated the effect of wild type PRMT3 or catalytically inactive mutant PRMT3-E338Q overexpression on the expression of RA responsive genes. We found that the overexpression of wild type PRMT3 or the catalytically inactive mutant PRMT3-E338Q significantly decreased the expression of all the tested RA responsive genes. Importantly, there was no significant difference in the impact of the wild type PRMT3 and PRMT3-E338Q mutant on the expression of tested RA responsive transcripts. This indicates that PRMT3 represses the expression of RA target genes in a methyltransferase activity independent manner.

We have added these findings in the **Supplementary Fig. 8** of the revised manuscript and have discussed this information in the ‘**PRMT3 negatively regulates ALDH1A1-mediated retinoic acid signaling**’ sub-section of the Results section of the revised manuscript (Page 9).

Collectively these data establish that PRMT3 negatively regulates the RA signalling in methyltransferase activity independent manner.

Comment 2.2: The section describing co-evolution of PRMT3 and ALDH1A1 is confusing, and do not contribute to the thesis of the manuscript. While the data does support a eukaryotic form of regulating ALDH1A1 activity, it is not clear these proteins reciprocally affected each other's evolution. This section should be removed (or moved to supplemental).

Response 2.2: The section on higher conservation of the C-terminal region of ALDH1A1 in species with PRMT3 (eukaryotes), compared to bacteria and archaea in which PRMT3 is absent, highlights the possibility that the C-terminal region of ALDH1A1 could have coevolved with the presence of PRMT3. This provides a complementary line of evidence of the importance of this interaction. Nevertheless, from the reviewer's suggestion, we learn that this section, when extensively discussed, could disrupt the flow of the manuscript. In view of this, we have moved the entire text and figure to the supplementary information (**Supplementary Fig. 2**) of the revised manuscript. We have highlighted the outcome of this analysis in one line in the subsection ‘**The catalytic domain of PRMT3 interacts with C-terminal region of ALDH1A1**’ of the Results section (Page 5).

Comment 2.3: On page 8, it is not clear in the following sentence what the fold changes are relative to ‘...down-regulation of PRMT3 means that ALDH1A1 would be in an unbound state, facilitating transcription of RA target genes, reflected by positive fold changes in the expression levels of RA target genes.’ Please clarify.

Response 2.3: We thank the reviewer for highlighting this point. To avoid confusion, we have now modified this sentence as ‘On the contrary, low levels of PRMT3 means that ALDH1A1 would be in an unbound state, facilitating transcription of RA target genes, reflected by positive fold changes in the expression levels of RA target genes’ in the revised manuscript (Page 10, paragraph 1). The fold changes for the RA targets were computed with the corresponding control samples and this information has been provided below and in the ‘**Transcriptome-wide analysis**’ subsection of Methods section (Page 18).

‘We obtained published datasets profiling differential expression of ALDH1A1 and PRMT3 from Expression Atlas. ALDH1A1 and PRMT3 were significantly differentially expressed in 133 and 54 conditions, respectively. We selected conditions in which (i) ALDH1A1 was up-regulated but PRMT3 was down-regulated ($AL1^{High}PR3^{Low}$) or (ii) both ALDH1A1 and PRMT3 were up-regulated ($AL1^{High}PR3^{High}$). Gene-expression changes in young cells (replicative) compared to spontaneously immortal (induced senescence) cells (Expression Atlas expression accession: E-GEOD-60340) showed significant up-regulation of ALDH1A1 and down-regulation of PRMT3 ($AL1^{High}PR3^{Low}$). Transcriptional changes in lung cancer compared to normal lung cells from freshly frozen human tissues (E-GEOD-19249), showed significant up-regulation of both ALDH1A1 and PRMT3 were significantly up-regulated ($AL1^{High}PR3^{High}$).’

Comment 2.4: The final section is highly speculative without genome-wide functional data. Here, again, use of a catalytically inactive PRMT3 mutant or PRMT3 inhibitor in the presence of RA, followed by microarray (or more extensive quantitative RT-PCR) would better support the authors’ model.

Response 2.4: We thank the reviewer for this suggestion. As mentioned in response 2.1, we investigated the effect of wild type PRMT3 or catalytically inactive mutant PRMT3-E338Q overexpression on the expression of RA responsive genes. We did not observe any significant difference between wild type PRMT3 and PRMT3-E338Q mutant on the reduction levels of tested RA responsive transcripts indicating that PRMT3 reduces the expression RA target genes in a methyltransferase activity independent manner. We have added this result in **Supplementary Fig. 8** of the revised manuscript and have provided this information in ‘**PRMT3 negatively regulates ALDH1A1-mediated retinoic acid signaling**’ sub-section of the Results section of the revised manuscript (Page 9).

As suggested by the reviewer, we have now increased number of RA responsive genes extensively (up to 13) in our quantitative RT-PCR analysis to better support the conclusions of the manuscript. We have added this result in **Fig. 6** of the revised manuscript and have provided this information in ‘**PRMT3 negatively regulates ALDH1A1-mediated retinoic acid signaling**’ sub-section of the Results section of the revised manuscript (Page 9).

Comment 2.5: Minor point, but the immunofluorescence in Figure 1F is correlative. While these images demonstrate that both proteins may be compartmentalized within the cytoplasm, the authors present numerous other (much stronger) data to support a PRMT3-ALDH1A1 interaction. These images may be moved to supplemental.

Response 2.5: As suggested by the reviewer, we have moved the immunofluorescence images to the **Supplementary Fig. 1** in the revised manuscript.

REVIEWERS' COMMENTS:

Reviewer #1 (Remarks to the Author):

In this revised submission, the authors have mostly addressed the previous reviewers' concerns. With added experimental data and analysis, the manuscript is in a much better shape. One minor suggestion is that Figure 4b illustrated that the interaction of PRMT3 inhibited the enzymatic activity of ALDH1A1. It would be valuable if the authors can provide a data curve to show PRMT3-concentration dependent inhibition of ALDH1A1 activity.

Point-by-point response to the reviewer comment

Reviewer #1

Comment 1.0: In this revised submission, the authors have mostly addressed the previous reviewers' concerns. With added experimental data and analysis, the manuscript is in a much better shape.

Response 1.0: We thank the reviewer for the positive evaluation of our manuscript.

Comment 1.1: One minor suggestion is that Figure 4b illustrated that the interaction of PRMT3 inhibited the enzymatic activity of ALDH1A1. It would be valuable if the authors can provide a data curve to show PRMT3-concentration dependent inhibition of ALDH1A1 activity.

Response 1.1: We would like to thank the reviewer for this suggestion. We have now performed the suggested experiment and found that the PRMT3 inhibits the enzymatic activity of ALDH1A1 in a concentration-dependent manner. We have added this data in Supplementary Fig. 3 of the revised manuscript and have discussed these results in 'PRMT3 inhibits the enzymatic activity of ALDH1A1' sub-section of the Results section of the revised manuscript (Page 7).